

**Spatiotemporal variations in terrestrial biospheric CO$_2$ fluxes of India derived from**
**MODIS, OCO-2 and TROPOMI satellite observations and a diagnostic terrestrial**
**vegetation model**
Aparnna Ravi [1,2], Dhanyalekshmi Pillai [1,2], Christoph Gerbig [3], Stephen Sitch[4], Sönke Zaehle[3],
Vishnu Thilakan[1,2], and Chandra Sekhar Jha [5]
Corresponding author: Dhanyalekshmi Pillai[1,2], dhanya@iiserb.ac.in
[1]Indian Institute of Science Education and Research Bhopal (IISERB), India,
[2]Max Planck Partner Group at IISERB, Bhopal, India,
[3]Max-Planck Institute of Biogeochemistry, Jena, Germany,
[4]University of Exeter, Exeter EX4 4QF, UK,
[5]National Remote Sensing Centre (ISRO), Balanagar, Hyderabad, India.



## Abstract

Accurate quantification of regional terrestrial fluxes is essential for improving our knowledge of the carbon sequestration potential of ecosystems, ecosystem functioning, and emission reduction demand in the context of climate change mitigation. However, the quantification is challenging owing to methodological and observational constraints, especially for regions with severe gaps in the ground-based observational network, like India. This study examines the potential of recent satellite missions, such as TROPOMI and OCO-2 providing retrievals of Solar-Induced chlorophyll Fluorescence (SIF) to improve terrestrial biosphere $CO_2$ flux estimates over India. Here, we present high-resolution estimates of Gross Primary Productivity (GPP) and Net Ecosystem Exchange (NEE) over India on a 0.1°×0.1° grid at a temporal resolution of 1 hour from 2012 to 2020. These products can be used for various applications such as those related to carbon cycle (e.g., inverse modelling of $CO_2$), benchmarking terrestrial biosphere models over the region, and understanding ecosystem responses to climate change. We follow a satellite-based diagnostic data-driven approach using a biosphere model, namely the Vegetation Photosynthesis and Respiration Model (VPRM) simulating both GPP and NEE, based on light use efficiency and satellite observations of the near-infrared radiance of vegetation (NIRv). We calibrate the standard VPRM GPP estimates using SIF-GPP relationship and investigate the model performance by comparing the simulations with eddy-covariance flux tower measurements. Our best model predictions are with a mean bias error (MBE) = 2.4 µmol m$^{-2}$ s$^{-1}$, root mean squared error (RMSE) = 3.8 µmol m$^{-2}$ s$^{-1}$ and squared correlation coefficient ($R^2$) = 0.56 when evaluating with observations at a monthly scale over the period from 2012 to 2018. The observed seasonal anomalies in NEE and GPP range from -4.9 to 8.0 µmol m$^{-2}$ s$^{-1}$ and -7.0 to 17.0 µmol m$^{-2}$ s$^{-1}$, respectively, and are well captured by our model. The model simulations are highly correlated with observations during 2018, the only common year when both EC and SIF observations are available,



with $R^2$ values of 0.68 and 0.74 for NEE and GPP, respectively. Incorporating the SIF signals in
the vegetation model improves model performance in capturing the seasonality and magnitudes of
GPP, thereby improving the estimates of NEE. We show the influence of soil temperature and soil
moisture on ecosystem respiration and refined the VPRM's ecosystem respiration calculation to
better constrain the fluxes, resulting in simulations closer to the observations. Ecosystem
respiration fluxes are less well constrained than ecosystem productivity fluxes due to the limited
observations. Based on satellite observations and the refined model, the annual NEE and GPP
estimates range from -0.38 Pg C yr$^{-1}$ to -0.53 Pg C yr$^{-1}$ (land C sink) and 3.39 Pg C yr$^{-1}$ to 3.88 Pg
C yr$^{-1}$, respectively over India for the years from 2012 to 2020. The biospheric flux distribution
over the region is found to be associated with ecosystem heterogeneity, and variations in
precipitation, and soil characteristics at a regional scale. Overall, our results show that the satellite-
based SIF data products can potentially inform the ecosystem-scale vegetation responses across
biomes over India. Future improvements in the terrestrial biosphere $CO_2$ flux estimates over India
can be attained through the carbon cycle data assimilation with the availability of both flux and
mixing ratio observations of $CO_2$.

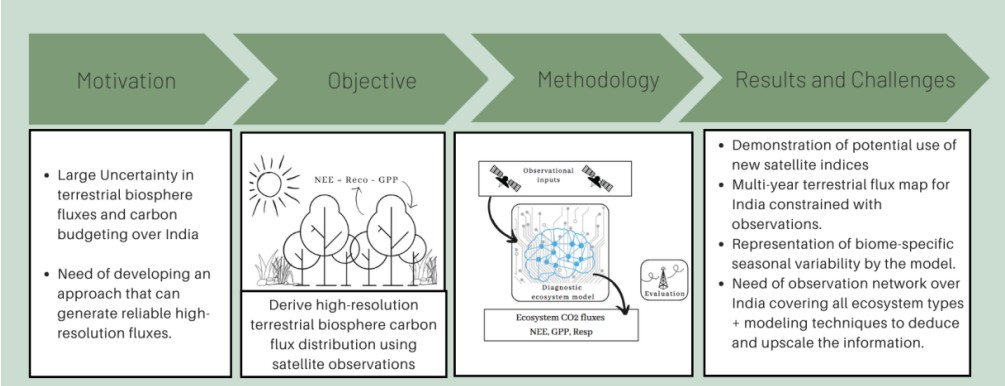



## 1. Introduction

The terrestrial biosphere is the largest sink of atmospheric $CO_2$. Globally, the net sequestration capacity of the terrestrial biosphere is ~3 Pg C yr$^{-1}$, corresponding to approximately a quarter of the global annual $CO_2$ emissions (Friedlingstein et al., 2022). Because of the vital role of the terrestrial biosphere in assimilating and exchanging atmospheric $CO_2$ with reservoirs, global initiatives to reduce greenhouse gas (GHG) emissions have included the active management of the terrestrial biosphere as a complementary measure for curtailing the emissions (Framework Convention on Climate Change available at http://www.unfccc.de/resource/cop3.htm) in the context of current and future climate.

However, the accurate estimation of terrestrial biosphere-atmosphere exchange fluxes at the scales relevant for climate change mitigation, which is well beyond the scale of single site observations, is still challenging. Major terrestrial fluxes, includes gross fluxes, Gross Primary Production (GPP), and Ecosystem respiration ($R_{eco}$), and their net, Net Ecosystem Exchange (NEE=$R_{eco}$-GPP), show considerable spatiotemporal variability owing to the differences in vegetation class and age, as well as in ecosystem response to the climate, geographic conditions, and other location-specific environmental factors (van der Meer et al., 2002). Terrestrial biosphere models can simulate these fluxes at different spatial and temporal scales over the globe (Peylin et al., 2013; Sitch et al., 2008, 2015; Thompson et al., 2016), however these model estimates often suffer from multiple sources of uncertainties, which include: the uneven distribution of eddy covariance flux tower observations worldwide for model validation or calibration, incomplete representation of vital processes in the model (e.g., drought-related mortality), and the insufficient understanding of how environmental factors affect atmosphere-biosphere carbon exchange. For example, the models are constrained with few observations over the Indian subcontinent, resulting in low confidence in the estimates of fluxes over India despite its important role in the global carbon



budget. The annual NEE estimates of India from previous studies range from 0 to -0.37 Pg C yr$^{-1}$
(Nayak et al., 2015; Patra et al., 2011; Rao et al., 2019). The spread among twelve vegetation
models in estimating the annual NEE of India for 2017 is 0.2 Pg C yr$^{-1}$, which is close to the
magnitude of the Indian terrestrial sink estimation itself (Sitch et al., 2015), leaving the country's
carbon flux estimates primarily uncertain.

Atmospheric $CO_2$ measurements, including those from satellite instruments, can be utilised

in an atmospheric inversion modelling framework to evaluate and improve the terrestrial biosphere
estimates of India. Simultaneously, prior estimates of biospheric fluxes with reasonable
spatiotemporal distributions are advantageous for the atmospheric inverse modelling to obtain the
optimal solution to the inverse problem with an improved confidence level (Michalak, 2004;
Rayner et al., 1999). The choice of prior and their spatiotemporal structures can be critical when
solving an ill-posed inverse problem (Rodgers, 2000). Previous studies have relied on the Light
Use Efficiency (LUE) model CASA (Carnegie Ames Stanford Approach;  Gamon et al. (1995))
and TRENDY model ensembles (Sitch et al., 2015) for estimating the spatiotemporal patterns of
biospheric $CO_2$ fluxes over southeast Asia covering India (Cervarich et al., 2016; Patra et al., 2011;
Peylin et al., 2013) and for India specifically (Goroshi et al., 2014; Nayak et al., 2010, 2013).
However, these models are employed at coarse resolution, e.g., $2' \times 2'$ spatial and monthly temporal
resolution for CASA, and TRENDY with sub-daily temporal resolution (with output available
monthly) and varying spatial resolution with respect to the model, typical 0.5° or above (see Table
3 for further details), with limited model validation against observations over India. This leads to
inadequate capturing of the spatiotemporal distribution of fluxes, resulting in varied estimates
among studies (Cervarich et al., 2016; Patra et al., 2013; Rao et al., 2019).



Recent advancements in satellite instruments, measuring Solar-Induced chlorophyll
Fluorescence (SIF) from space can be helpful, especially for the region with severe gaps in ground-
based in-situ observations. These satellite-based SIF retrievals, representing re-emitted solar
radiation at the long wavelength range (650–850 nm) by the chlorophyll-a pigment, can be utilised
to improve the prior estimates of carbon uptake through photosynthesis at regional to global scales
(Frankenberg et al., 2011; Gu et al., 2019; Köhler et al., 2018; Li et al., 2018; Smith et al., 2018;
Sun et al., 2017; Yu et al., 2019). Since the re-emission process (fluorescence) by chlorophyll is
linked to the primary steps in photosynthesis, SIF can be used as the proxy for photosynthesis
(Parazoo et al., 2018; Sun et al., 2018; Yu et al., 2019). Only ~2% of the incident solar energy
absorbed by green plants is re-emitted by chlorophyll as fluorescence. Thus, SIF retrievals from
space need advanced spectrometers with a high spectral resolution and a high Signal-to-Noise Ratio
(SNR) due to narrow Fraunhofer lines and weak signals. However, SIF observations are prone to
systematic errors which are associated with the strength and extraction range of the signal (Joiner
et al., 2016; Köhler et al., 2015; Li et al., 2018). The SIF-GPP relationship can become weak in
certain environmental conditions such as drought (e.g., Shekhar et al. (2022) and variable within
certain biome based on leaf physiology (e.g., Wu et al. (2022)). The first satellite-based global
retrievals of SIF are achieved by the Fourier transform spectrometer (fluorescence spectrum at
755–775 nm) on board the Greenhouse gases Observing SATellite (GOSAT). Other satellite
missions that provide SIF retrievals at different spatial and temporal resolutions are GOME-2
(Global Ozone Monitoring Experiment 2; Frankenberg et al. (2011)), OCO-2 (Orbiting Carbon
Observatory 2; Sun et al. (2018)), OCO-3 (Orbiting Carbon Observatory 3; Taylor et al. (2020)),
and TROPOMI (TROPOspheric Monitoring Instrument; Guanter et al. (2021)).
This study presents high-resolution terrestrial biosphere $CO_2$ flux estimates over India on a
0.1°×0.1° grid at a temporal resolution of 1 hour for the period from 2012 to 2020. These high-





resolution biospheric flux products can be used in the near-future as prior estimates in the inverse
data assimilation of $CO_2$ or can be coupled with high-resolution transport models for understanding
the atmospheric $CO_2$ transport or variability associated with natural fluxes. We follow a diagnostic
data-driven approach using a biosphere model based on light-use efficiency and satellite
observations of SIF and demonstrate their potential to capture the spatiotemporal variations of
biosphere fluxes. The gridded NEE, GPP and $R_{eco}$ are initially generated by utilising the diagnostic
satellite-based biosphere model, namely Vegetation Photosynthesis and Respiration Model
(VPRM; Mahadevan et al. (2008)). Previously, Thilakan et al. (2022) have generated the VPRM
simulations of terrestrial biosphere fluxes (NEE, GPP, and $R_{eco}$) over the Indian subcontinent at a
spatial resolution of $0.1°×0.1°$ and a temporal resolution of 1 hour using uncalibrated model
parameters. These VPRM fields are revised by improving the ecosystem uptake across different
biomes using SIF retrievals from OCO-2 and TROPOMI, which provide much finer resolutions
and higher data density over the region than those from previous missions (e.g., GOSAT and
GOME-2). As we expect a distinct contribution of soil moisture stress in ecosystem respiration
signals, we also re-define $R_{eco}$ calculation in the VPRM (originally as a linear function of air
temperature) to include the influence of both, soil temperature and soil moisture so that the NEE
estimates can be improved. A recent study over the Eastern USA and Canada has also showed
improvements in $R_{eco}$ simulations when including the influence of changing foliage, water stress
and non-linear dependence of temperature (Gourdji et al., (2022).
Variations in temperature, radiation, and resource availability (e.g., water and soil nutrients)
influence plant phenology and ecosystem stress levels, contributing to seasonal anomalies in GPP
and NEE. It remains challenging to accurately represent the seasonal dynamic attributes of
ecosystem fluxes and simulate their associated variability. In this study, we assess the usefulness
of the SIF signals to capture the seasonality and magnitudes of GPP in the model by comparing



them with eddy-covariance flux tower measurements from India for the period from 2012 to 2018.
We further investigated the influence of environmental factors and processes on modelled
respiration at the regional level. We assess the VPRM against estimates from TRENDY model
ensemble and Carbon Tracker inversion. By improving the diagnostic biospheric model and
generating simulations at a high resolution, comparing the derived flux components from multiple
terrestrial models, and evaluating the improved model against observations, we investigate the
spatial and temporal variations of biosphere fluxes in different ecosystems over India on seasonal
and annual scales.
**2. Methods**
For deriving improved estimates of terrestrial biosphere $CO_2$ fluxes across the ecosystem
over India: i) we implement and customise the standard VPRM for a domain covering India (5°N
to 40°N, 66°E to 100°E, Fig. 1 and Fig. S1) and perform the simulations of NEE, GPP and $R_{eco}$
fluxes (Sect. 2.1); ii) we derive ecosystem-specific linear relations between SIF and GPP using SIF
retrievals based on OCO-2 and TROPOMI (detailed in Sect. 2.2); iii) we apply the above satellite-
derived information in the VPRM to improve the estimates of the ecosystem uptake (Sect. 2.2);
and iv) we further modify the VPRM-derived ecosystem respiration to include the influence of soil
temperature and soil moisture specific to vegetation classes (Sect. 2.3).
We compare the standard and improved VPRM simulations with the TRENDY model
ensemble and other model simulations (Sect. 2.4) and evaluate the simulations with the flux tower
observations (Sect. 2.5). In this section, we also describe the approaches used for overall analyses
for assessing the model's performance and deriving the spatiotemporal characteristics of fluxes
(Sect. 2.6). An overview of the datasets used in the study is presented in Table 1.
**2.1 VPRM model implementation**



The standard VPRM employs a remote sensing-based scheme to obtain high-resolution
estimates of NEE, GPP and $R_{eco}$, using Enhanced Vegetation Index (EVI) and Land Surface Water
Index (LSWI), derived from the Moderate Resolution Imaging Spectroradiometer (MODIS)
measurements onboard the NASA's Terra and Aqua satellites. We use the MODIS tiles of the
surface reflectance dataset (MOD09A1) on sinusoidal grids at a 500 m spatial resolution with an
8-day interval to generate EVI and LSWI fields. Specifically, we use the red band (band 1), the
near-infrared band (band 2), the blue band (band 3) for deriving EVI, and the near-infrared band
(band 2) and the shortwave infrared band (band 6) for deriving LSWI. For representing different
biomes in VPRM, we use vegetation classification based on SYNMAP (Jung et al., 2006).
In VPRM, NEE for each vegetation class is calculated based on GPP (light-dependent term)
and $R_{eco}$ (light-independent term). NEE is assessed based on the sign convention where negative
values indicate $CO_2$ uptake and positive values represent $CO_2$ release into the atmosphere.
$$NEE = -GPP + R_{eco} \tag{1}$$
$$GPP = \lambda \times P_{scale} \times W_{scale} \times FPAR_{PAV} \times \frac{1}{[1+(SW_{down}/SW_{down0})]} \times SW_{down} \times T_{scale} \tag{2}$$
$$R_{eco} = \alpha \times T_{air} + \beta \tag{3}$$
where $\lambda$ is the factor representing light use efficiency. $FPAR_{PAV}$ is the fraction of
photosynthetically active radiation available to the photosynthetically active part of vegetation
which is derived from MODIS EVI. $T_{scale}$, $P_{scale}$ and $W_{scale}$ are dimensionless scalars representing
the sensitivity of plants to changes in temperature, phenology, and water availability, respectively.
$T_{scale}$ is derived using ecosystem-specific temperature as follows:
$$T_{scale} = \frac{(T - T_{min})\,(T - T_{max})}{(T - T_{min})\,(T - T_{max}) - (T - T_{opt})^2} \tag{4}$$
where $T_{opt}$, $T_{max}$, $T_{min}$ represent optimal, maximum, and minimum temperatures for photosynthesis
activity for each vegetation class. Photosynthesis is assumed to be absent above or below $T_{max}$ and





$T_{min}$, respectively. $T_{air}$ is the hourly air temperature at 2 m prescribed from ERA5 (Dee et al., 2011).
In this study, we set $T_{opt}$, $T_{min}$ and $T_{max}$ to 20 °C, 0 °C and 45 °C, respectively. We utilise $P_{scale}$ to
account for the effects of leaf age on photosynthesis; hence it is set to 0 for water bodies and
unclassified vegetation classes. $P_{scale}$ is assumed to always be 1 for the Evergreen vegetation class.
For all vegetation classes other than Evergreen, we compute $P_{scale}$ as a function of LSWI except at
the time of maximum greenness (representing full leaf expansion) as follows:
$$P_{scale} = \frac{1+LSWI}{2}$$    (5)
For the maximum greenness time, $P_{scale}$ is set to 1.
$W_{scale}$ is used to represent the effect of water stress on photosynthesis and is derived as follows:
$$W_{scale} = \frac{1+LSWI}{1+LSWI_{max}}$$    (6)
PAR is the photosynthetically Active Radiation, which is calculated based on incoming shortwave
solar radiation ($SW_{down}$; µmol m$^{-2}$ s$^{-1}$). $SW_{down}$ is prescribed from ERA5.

In Eq. (3), $T_{air}$ is constrained with a threshold value ($T_{tshld}$), and $T_{air}$ below $T_{tshld}$ is set to

$T_{tshld}$ for accounting for ecosystem respiration in winter times. Negative values of $R_{eco}$ are set to 0.

The VPRM parameters, $\lambda$, $SW_{down0}$, $\alpha$, and $\beta$ are usually calibrated against site-level eddy

covariance measurements across different ecosystem types by minimising the least squares
between VPRM fluxes and eddy flux tower observations. This optimization procedure with discrete
tower locations representing major vegetation classes is expected to enhance the model
performance for the region of interest (Dayalu et al., 2018; Luus & Lin, 2015). Due to the lack of
availability of sufficient observational eddy flux measurements for calibration for India, we use the
VPRM parameters that were originally optimised against the Amazonian Tropical biomes (Botía
et al., 2022) but modified as given in Table 2. We acknowledge that these parameters are not



necessarily representing subtropical Indian biomes, which may lead to reduced model performance
compared to other VPRM model simulations for regions like Europe or North America.
**2.2 Ecosystem uptake refinements using SIF**

As the reliability of the standard VPRM simulations depends on the model parameters,

which are currently not specific to Indian biomes, we use satellite products based on OCO-2 and
TROPOMI deriving the relationships between SIF and GPP across different vegetation classes and
utilise them to improve the VPRM estimates of GPP.

We use two SIF products: GOSIF_v2 (http://data.globalecology.unh.edu/; Li & Xiao

(2019a)), and the TROPOMI based product TROPOSIF (http://ftp.sron.nl/open-access-data-
2/TROPOMI/tropomi/sif/v2.1/l2b/; Köhler et al. (2018)). GOSIF_v2 (hereafter referred to as
GOSIF) provides SIF retrievals at spatial and temporal resolutions of 0.05° and 8-day. The spatial
discontinuity in the original daily OCO-2 retrievals is improved in GOSIF using a machine learning
approach based on MERRA-2 meteorological fields, MODIS reflectance and landcover data,
preserving the observed variability of discrete SIF retrievals, as explained in (Li & Xiao, 2019a).
In addition to SIF products, we also use the GPP product derived from OCO-2 SIF (Li & Xiao,
2019b), namely GOSIF_GPP_v2, providing 8-day GPP at 0.05° grid resolution for model
comparison (see details below). Hourly SIF retrievals are available from TROPOMI (hereafter
referred to as TROPOSIF) at 0.1° spatial resolution from May 2018 onwards.

We assumed GPP$_{SIF}$ (i.e., GPP derived from SIF) to be varied linearly with SIF (Sun et al.,

2017; Zhang et al., 2016). The SIF-GPP relationship across the vegetation classes in VPRM is
derived as follows:
$GPP_{SIF}(vg) = \gamma_{vg} \times SIF_{vg} + C_{vg}$                                         (7)



Here $\gamma_{vg}$ is the factor converting SIF to GPP and $C_{vg}$ represents the constant, specific to each biome
$vg$. The biome specific $\gamma_{vg}$ and $C_{vg}$ over India are derived from the 8 day averaged OCO-2 derived
GPP (GOSIF_GPP_v2) and SIF (GOSIF) products that followed the optimization procedure as
described in Li & Xiao, (2019b), which are separated for each vegetation classes, denoted as
$GPP_{OCO2}(vg)$ and $SIF_{OCO2}(vg)$. $\gamma_{vg}$ and $C_{vg}$ are thus the linear slope between $GPP_{OCO2}(vg)$ and
$SIF_{OCO2}(vg)$, and the y-intercept respectively. When using TROPOSIF, the factor of difference
between GOSIF and TROPOSIF values ($S_{GOSIF}(vg)$) is taken in to account to derive SIF-GPP
relationship: i.e., $\gamma_{TROPOSIF,vg} = \gamma_{vg}/S_{GOSIF}(vg)$ and $C_{TROPOSIF,vg} = C_{vg}/S_{GOSIF}(vg)$ (see Sect.
3.1 for more details).
The distribution of GPP derived by the VPRM ($GPP_{vprm,STD}$) is improved by up-scaling it as
follows:
$$GPP_{vprm,mod}(i,j,t,vg) = \eta_{vg} \times GPP_{vprm,STD}(i,j,t,vg) + \varepsilon \qquad (8)$$
i, j, and t correspond to latitude, longitude, and time respectively. $\eta_{vg}$ is the scaling factor
corresponding to the specific vegetation class, applied to upscale $GPP_{vprm,STD}$ to include the
information provided by SIF. $\eta_{vg}$ is thus:
$$\eta_{vg} = \frac{\Sigma(GPP_{SIF}(vg) \times GPP_{vprm,STD}(vg))}{\Sigma GPP_{vprm,STD(vg)}^2} \qquad (9)$$
**2.3 Soil moisture and temperature in respiration model equation**
The soil properties can influence both autotrophic and heterotrophic respiration, especially
over a region with distinct wet and dry seasons (Flexas et al., 2006; Meir et al., 2008; Molchanov,
2009). Since the standard VPRM constructs ecosystem respiration as a simple linear function of
air temperature, here we assess the impact of soil temperature and soil moisture (SM/ST) content
in ecosystem respiration and refine the formulation accordingly. We utilise the SM/ST fields from



the high-resolution land data assimilation system (HRLDAS; Chen et al. (2007)) based on the Noah
land surface model (LSM), providing 3 hourly fields at 4 km spatial resolution for the period 2012
to 2017. As this data product does not cover our analysis period, we also use the SM fields from
GLEAM v3 (https://www.gleam.eu/#datasets; Martens et al. (2017)) model and ST from ERA5
(https://cds.climate.copernicus.eu/cdsapp#!/dataset/reanalysis-era5-land?tab=overview; Hersbach
et al. (2020)) reanalysis product (see Table 1).
The distribution of $R_{eco}$ derived by the standard VPRM is re-defined as follows:
$R_{eco,vprm,mod}(i,j,vg) = T_{s,vg}.ST(i,j,vg) + M_{s,vg}.SM(i,j,vg) + R_{vg}.\big(\alpha_{vg}.T_{air}(i,j,vg) +$
$\beta_{vg}\big)$ (10)
where, $T_{s,vg}$, $M_{s,vg}$ and $R_{vg}$ represent the vegetation specific parameters derived using the multi-
linear regression with soil temperature (ST), soil moisture (SM), and standard VPRM respiration
against observation-based respiration fluxes. Here, we used two available observation-based
datasets to calibrate respiration model parameters. The terrestrial vegetation fluxes (specifically
ecosystem respiration fluxes) derived from 1) FLUXNET
(https://db.cger.nies.go.jp/DL/10.17595/20200227.001.html.en, see Table 1, Zeng, Jiye (2020))
and 2) FLUXCOM (https://www.bgc-jena.mpg.de/geodb/projects/DataDnld.php, see Table 1,
Jung et al. (2020)) observational database are used for parameter optimization. Table 2 provide the
details of the vegetation specific model parameters derived for refining $R_{eco}$.
**2.4 Other model products for comparison**
For the inter-model comparison and performance assessment, we use simulated surface
$CO_2$ fluxes from process-based terrestrial biosphere models commonly used for carbon cycle
studies and the global inverse modelling system providing flux estimates consistent with
atmospheric mixing ratio observations.



We have used process-based simulations generated by 14 Dynamic Global Vegetation
Models (DGVM's) employed in the TRENDYv10 model ensemble for the Indian region (see Table
3). All land surface models under TRENDY were driven with common input/forcing data from
1901 to 2020 and followed a common simulation protocol. Model simulations include climate
forcing from CRU+CRU-JRA (https://crudata.uea.ac.uk/cru/data/hrg/cru_ts_4.05/) monthly and 6
hourly historical forcing for the period 1901 to 2020, ice core data from 1700 to 2020 and land-use
change data from Hyde database for the period 850 to 2021. Specifically, this study uses TRENDY
S3 simulation products, which consider the impact of atmospheric $CO_2$ concentration changes,
climate change, and land cover changes on the global terrestrial ecosystem GPP (see
https://blogs.exeter.ac.uk/trendy/). The TRENDY models used in this study differ in spatial
resolution, but each provides fluxes at a monthly temporal resolution.
We use inverse model estimates of fluxes provided by the Carbon Tracker (CT2019B,
hereafter referred to as CT) modelling system
(https://gml.noaa.gov/ccgg/carbontracker/download.php; Peters et al. (2007)). The prior fluxes for
the biospheric module of CT were from a diagnostic CASA biogeochemical model based on the
remote-sensed monthly fraction of Photosynthetically Active Radiation (fPAR). Three hourly
gridded estimates of optimised biospheric $CO_2$ fluxes with a horizontal resolution of $1° \times 1°$ over
the Indian domain for the years 2016 to March 2019, available at
https://gml.noaa.gov/ccgg/carbontracker/ are used in this study.
All these gridded flux estimates used for comparing spatial patterns are aggregated or
disaggregated to a common spatial and monthly temporal resolution for comparison (see Sect. 2.6).
**2.5 EC flux tower observations for model evaluation**
For the model evaluation, we use eddy covariance observations of terrestrial biosphere $CO_2$
fluxes from a flux tower located at Betul (21°51'46.84'' N latitude and 77°25'33.67'' E longitude,



Madhya Pradesh; Jha et al. (2013)) in the Central Indian state of Madhya Pradesh. Betul tower
(commissioned in November 2011) is 507 m above mean sea level inside the mixed Deciduous
forest where a tropical climate prevails. Further descriptions of the site and details of the
instrumentation from Betul can be found in (Jha et al., 2013; Rodda et al., 2021). Table 4 provides
an overview of the characteristics of the flux tower site, and Fig. 1 shows the location map of the
flux towers under this study.
The half-hourly data from Betul is aggregated into hourly, daily, monthly and annual time
scales for this analysis. All the available data from 2012 to July 2019 is used in this study (more
details can be seen in Rodda et al. (2021)). There exist data gaps for specific years. For the
evaluation analyses, model simulations are compared to observations at hourly, daily and monthly
timescales. We estimate mean biases error (MBE), root mean squared error (RMSE), and squared
correlation coefficient ($R^2$) to assess the model's efficiency in predicting the magnitude and
variability.
**2.6 Spatial and Biome-specific Pattern analysis**
Here, we use flux simulations generated by refined VPRM, TRENDY model ensemble and
CT, re-gridded to a spatial resolution of $1° \times 1°$, to examine spatial gradients and seasonal variations
of biospheric fluxes. Since some ecosystems can be more biologically productive than others, we
aggregated flux patterns separately for each vegetation class based on SYNMAP land cover types
for estimating each ecosystem's productivity in capturing atmospheric $CO_2$. We have also
considered different periods, such as pre-monsoon (March to May), monsoon (June to September)
and post-monsoon (October to December), to assess the seasonally varying biome productivity.
We use improved VPRM fluxes at hourly time scales for these ecosystem-based analyses.
**3. Results and Discussion**





### 3.1 Spatial and temporal patterns of SIF over Indian biomes


As explained in Sect. 2.2, we utilise satellite retrievals of SIF from OCO-2 (GOSIF) and

TROPOMI (TROPOSIF) to improve VPRM-derived GPP ($GPP_{vprm,STD}$). Here, we present biome-
specific analyses of SIF products, deducing their spatial and temporal characteristics over Indian
biomes from 2018 to 2020. For the spatial analysis, the monthly and annual mean GOSIF data have
been regirdded to 0.1°✕0.1°. Both 8-day averaged SIF products agree with each other across
biomes with $R^2$ ranging from 0.45 to 0.62 except for Grassland ($R^2 = 0.22$) (see Table S1). A similar
good agreement between SIF retrievals from OCO-2 and TROPOMI on global scale is also
reported by Köhler et al. (2018) and Guanter et al. (2021).

Annually, the highest SIF values (GOSIF, mean/min/max: 0.28/0.03/0.44 mW m$^{-2}$ sr$^{-1}$ nm$^{-1}$

and TROPOSIF, mean/min./max: 1.18/0.17/1.93 mW m$^{-2}$ sr$^{-1}$ nm$^{-1}$ for the year 2019) are
exhibited by Evergreen forest, and the lowest values are observed (GOSIF, mean/min/max:
0.07/0/0.24 mW m$^{-2}$ sr$^{-1}$ nm$^{-1}$, TROPOSIF, mean/min/max: 0.41/0/1.61 mW m$^{-2}$ sr$^{-1}$ nm$^{-1}$) over the
desert regions of Rajasthan where Shrubland vegetation dominates. Over the years (2019 to 2020),
based on GOSIF, the rates of an annual increase in SIF value for Cropland, Savanna, Shrubland,
Deciduous forest, and Evergreen forest are in the range of 0.01 mW m$^{-2}$ sr$^{-1}$ nm$^{-1}$ to 0.23 mW m$^{-2}$
sr$^{-1}$ nm$^{-1}$, with Grassland showing no enhancement. Mixed Forest biomes exhibit a negative growth
rate of -0.005 mW m$^{-2}$ sr$^{-1}$ nm$^{-1}$. Like GOSIF, TROPOSIF also indicates zero growth rate for
Grasslands, while other ecosystems show an annual growth rate between 0.04 mW m$^{-2}$ sr$^{-1}$ nm$^{-1}$ to
0.11 mW m$^{-2}$ sr$^{-1}$ nm$^{-1}$. On an annual scale, large spatial variability in the SIF values is exhibited
by Shrubland and the least by Savanna. Overall, we find that TROPOSIF values (based on SIF
retrievals at 735 nm) are ~4 times greater than GOSIF (based on SIF retrievals at 757 nm) over the
study region for all the biomes except for Grassland, where the biome-specific TROPOSIF is ~3





times larger than GOSIF. Hence, we scaled up GOSIF and the derived scaling factors are specific
to each biome (see Table S1). A similar up scaling of OCO-2 SIF is also done by Köhler et al. (
2018) and Guanter et al. (2021) for comparing the fields with TROPOSIF on a global scale. In Fig.
2, we compare scaled GOSIF and TROPOSIF across different biomes.
We find that the spatial heterogeneity observed in SIF emission is directly related to the
vegetation class and the availability of rainfall. For example, biomes in Central, North East and
South West India, where significant rainfall occurs during the summer monsoon period (June -
August), show higher fluorescence than the rest of the region (see Fig. 3). All vegetation classes
exhibit large seasonal variability with a seasonal maximum from June to July and a seasonal
minimum from March to April (see Fig. 4), indicating changes in the rate of photosynthesis with
rainfall availability with correlation values ranging from 0.78 to 0.93. A similar high positive
correlation between precipitation and SIF is indicated by Albright et al. (2022) over the Amazon
region. No significant influence of rainfall is found in the seasonality over Grassland ($R^2$ = <0.4).
Cropland and Shrubland vegetation show the primary maximum with the onset of monsoon (June-
July) and the secondary maximum during winter months (January-February). These two seasonal
maxima are consistent with the prominent crop-growing seasons of India (Nayak et al., 2010),
which are associated with enhanced primary productivity. Compared to GOSIF, TROPOSIF better
exhibits the double peak in SIF temporal distribution for both ecosystems over this region.
**3.2 SIF-GPP relationship across different biomes**
We have derived SIF-GPP relationship similar to Li & Xiao (2019b) using up scaled GOSIF
and $GPP_{SIF}$ across different biomes over India, as mentioned in Sect. 2.2 (see Table 5). Li & Xiao
(2019b) used linear relationship between GOSIF flux tower network of observations (FLUXNET;
Baldocchi et al. (2001)) based GPP to map GPP globally. Our derived scalars for converting SIF
to GPP are different from Li & Xiao (2019b) due to the differences in Indian biomes, their





classifications, and the up-scaling of the GOSIF product (see Table 5). The derived scalars for
converting SIF to GPP range from 4.80 to 7.84 mW m$^{-2}$ sr$^{-1}$ nm$^{-1}$/μmol m$^{-2}$ s$^{-1}$ for different biomes.
While both SIF patterns are in good agreement with VPRM-derived GPP over most of the
vegetation classes under our study (e.g., $R^2$ = 0.77 to 0.85 for Shrubland), we find a weak
correlation between SIFs and standard VPRM-derived GPP for Savanna ($R^2$ = 0.09 to 0.36). The
above correlation values are based on the annually averaged data analysis from 2018 to 2019 (not
shown).
**3.3 Model evaluation with eddy covariance flux observations**
Figure 5 shows the inter-annual variations in monthly averaged fluxes of GPP, R$_{eco}$, and
NEE over Betul from 2012 to 2018. A significant data gap exists during 2014 and 2017. Since
Betul is a tropical Deciduous forest, the strong seasonality exhibited by the observed fluxes can be
associated with changes in plant physiology throughout the year. Based on Betul observations,
Rodda et al. (2021) reports a net sink at site level with an annual NEE, GPP and R$_{eco}$ of -524 ± 40
g C m$^{-2}$ yr$^{-1}$, 3358 ± 167 g C m$^{-2}$ yr$^{-1}$, and 2834 ± 157 g C m$^{-2}$ yr$^{-1}$, respectively. While observed
NEE shows positive values (representing carbon release to the atmosphere) during summer (March
- June), the ecosystem uptake was observed (negative NEE values) for the rest of the year (July -
February). Seasonal maxima for GPP range from 19 μmol m$^{-2}$ s$^{-1}$ to 25 μmol m$^{-2}$ s$^{-1}$ from July to
September due to peak photosynthetic activity associated with optimal water and moisture
availability. The forest site receives rain from June onwards, with maximum precipitation during
July (South West monsoon period, based on TRMM precipitation data). However, the ecosystem
productivity is less in June due to a shortage in photosynthetically active solar radiation owing to
cloud cover, as seen from satellite images (https://www.mosdac.gov.in/). Also, the transition in
vegetation development from dry summer to wet periods occurs during the early monsoon month
(June). The availability of rainfall and radiation enhances plant productivity at the site, Rodda et



al. (2021) noted. The variability in seasonal maxima over the year can thus be associated with the
inter-annual variability of the summer monsoon. Ecosystem productivity reaches its annual
minimum during March and April (1 $\mu$mol m$^{-2}$ s$^{-1}$ to 3 $\mu$mol m$^{-2}$ s$^{-1}$) due to the leaf shedding of
Deciduous vegetation during summer. Ecosystem respiration showed two peaks, a primary peak
during early monsoon months (June & July) and a secondary peak during late monsoon months
(August & September). These respiration peaks are associated with increased air temperature when
autotrophic respiration is expected to increase and enhanced soil microbial respiration when
attaining sufficient soil moisture. An increase in vegetation greenness with water availability also
enhances autotrophic respiration. A sharp fall in $R_{eco}$ after the primary maxima can likely be due
to the decrease in soil respiration due to water logging associated with enhanced precipitation
creating anoxic conditions and limiting microbial activity in the area (Han et al., 2018). The
conditions become favourable for autotrophic and heterotrophic respiration during post-monsoon
(enhanced vegetation greenness and optimal soil moisture content), resulting in the observed
secondary maximum. We find weak ecosystem respiration from November to May (2 $\mu$mol m$^{-2}$ s$^{-}$
$^{1}$ to 7 $\mu$mol m$^{-2}$ s$^{-1}$) owing to the leaves shedding and reduced soil respiration, limited by dry soil.

On comparing observations with model simulations, standard VPRM (hereafter referred to

as VPRM$_{STD}$) shows better agreement in predicting the seasonality in observed monthly averaged
NEE fluxes ($R^2$ = 0.59) than CT ($R^2$ = 0.24) and TRENDY ($R^2$ = 0.45), but with a significant
underestimation of NEE fluxes at a monthly scale (see Table 6).  The model bias increases from
August to December (MBE = 4.83 $\mu$mol m$^{-2}$ s$^{-1}$ and RMSE = 5.0 $\mu$mol m$^{-2}$ s$^{-1}$) compared to other
periods. Note that we have used the TRENDY model ensemble for the comparison, and the
variation among TRENDY model simulations for NEE (as calculated by the standard deviation
from the ensemble mean over the seven years) ranges from -2.84 $\mu$mol m$^{-2}$ s$^{-1}$ to 1.80 $\mu$mol m$^{-2}$ s$^{-}$
$^{1}$ over Betul. Similar to NEE, the model predicted the monthly mean variations in GPP reasonably



well ($R^2$ = 0.71), but with considerable bias (MBE = -6.7 µmol m$^{-2}$ s$^{-1}$, RMSE = 8.3 µmol m$^{-2}$ s$^{-1}$).
The model-observation bias for GPP is found to be high during productive months (June-
December). Previous studies have shown the underestimation of GPP when MODIS-derived
products are used for GPP estimation (e.g., Zhang et al., 2008). The GPP underestimation by
VPRM$_{STD}$ can be thus related to the usage of MODIS reflectance products. Overall, VPRM$_{STD}$
captures the seasonal pattern in NEE and GPP compared to other biospheric models with different
model structures, such as the inversion product CT and the ensemble of process-based models
TRENDY.

We further investigated reducing the model-observation bias in the VPRM$_{STD}$ model. In

addition to standard datasets in VPRM$_{STD}$, we utilised GPP$_{SIF}$ products, soil moisture and soil
temperature to improve GPP and R$_{eco}$ simulations. Incorporating SIF in simulating the VPRM GPP
has noticeably improved the ability of the model to capture the observed seasonal variability (see
Fig. 5). Both GPP$_{GOSIF}$ and GPP$_{TROPOSIF}$ show good agreement in capturing the seasonal variations
($R^2$ = 0.65 to 0.68), with values closer to the observation. Though SIF based GPP products are
closer than $GPP_{vprm,STD}$ to the observed GPP in terms of magnitude, the observed patterns in GPP
are better captured by VPRM$_{STD}$ ($R^2$ >0.7) than other products (see Sect. 3.3). This shows the
potential of VPRM model to predict the observed variations in GPP, leading to calibrate VPRM
model parameters rather simply using GPP$_{GOSIF}$ and GPP$_{TROPOSIF}$ in our NEE estimations. VPRM
GPP modified based on GOSIF (hereafter referred to as VPRM$_{GOSIF}$), and VPRM modified based
on TROPOSIF (hereafter referred to as VPRM$_{TROPOSIF}$) are evaluated with observations, and the
inter-comparison with VPRM$_{STD}$ shows remarkable improvement in the model performance for
GPP with a significant reduction in RMSE and MBE values (see Fig. 5a and Table 6). For GPP,
the bias reduced significantly for refined models (RMSE: VPRM$_{GOSIF}$ = 4.9 µmol m$^{-2}$ s$^{-1}$, and



$VPRM_{TROPOSIF}$ = 4.3 µmol m$^{-2}$ s$^{-1}$ and MBE: $VPRM_{GOSIF}$ = -3.3 µmol m$^{-2}$ s$^{-1}$, $VPRM_{TROPOSIF}$ = -2.6 µmol
m$^{-2}$ s$^{-1}$). The observed seasonal anomalies in GPP (variability after subtracting the decadal mean),
associated with ecosystem stress and phenology, ranges from -7.0 to 17.0 µmol m$^{-2}$ s$^{-1}$ with a
standard deviation of 6.3 µmol m$^{-2}$ s$^{-1}$. These variations are well captured by our model with a
mean bias of -1.8 µmol m$^{-2}$ s$^{-1}$. The above levels of model improvements confirm the potential of
using high-resolution satellite-derived SIF in capturing the seasonal cycle of GPP at an ecosystem
level. Hence, our results are broadly consistent with Qiu et al. (2020); Joiner et al. (2018); and
Wood et al. (2017). As a direct proxy for photosynthesis, SIF is expected to provide improved
estimates than conventional vegetation indices (Zhang et al., 2016) (e.g., EVI, LSWI) used in
VPRM GPP estimation.

The $VPRM_{STD}$ model fails to capture the seasonality in respiratory fluxes ($R^2$ = 0.02) for

the period from 2012 to 2018, with a significant underestimation of ecosystem respiration by -3.5
µmol m$^{-2}$ s$^{-1}$ (RMSE values: ~5.7 µmol m$^{-2}$ s$^{-1}$). To improve the model performance, we performed
three sets of modified VPRM simulations for $R_{eco}$, utilising observation-based datasets in addition
to those already used for $VPRM_{STD}$ $R_{eco}$ simulations, such as 1. ST, 2. SM, and 3. both ST and SM.
$R_{eco}$ modified based on various datasets (e.g., HRLDAS ST/SM, ERA5 ST, and GLEAM SM)
provide similar results. Here we present the analysis using ERA5 ST and GLEAM SM, considering
the large temporal coverage of the data. VPRM respiration modified using SM (Fig. 5b) shows
much improvement in model prediction ($R^2$: 0.80) than when ST alone is used. VPRM respiration
modified using both SM and ST (i.e., $VPRM_{MOD}$) shows slightly better improvement than using
only SM. The model-observation bias reduced considerably, with RMSE reducing from 5.7 µmol
m$^{-2}$ s$^{-1}$ to 1.9 µmol m$^{-2}$ s$^{-1}$ and MBE reducing from -3.5 µmol m$^{-2}$ s$^{-1}$ to -0.01 µmol m$^{-2}$ s$^{-1}$. In general,
incorporating the soil temperature and soil moisture in addition to air temperature in the ecosystem
respiration calculation in the VPRM improves the model's ability to simulate more realistic values





over the Deciduous ecosystem of Betul. The improvement in VPRM $R_{eco}$ while incorporating soil
temperature is also reported elsewhere (e.g., Luus et al., 2015).
The VPRM NEE estimated based on modified GPP from VPRM$_{GOSIF}$ and $R_{eco}$ from
VPRM$_{MOD}$ (hereafter referred to as VPRM$_{GOSIF,SMST}$) and based on VPRM$_{TROPOSIF}$ and VPRM$_{MOD}$
(hereafter referred to as VPRM$_{TROPOSIF,SMST}$) are evaluated with observation over Betul (Fig. 5c).
The modified models showed improvement over VPRM$_{STD}$ in capturing the observed seasonal
pattern with a reduction in errors during the period from 2012 to 2018 (RMSE: VPRM$_{GOSIF,SMST}$ =
4.4 µmol m$^{-2}$ s$^{-1}$, VPRM$_{TROPOSIF,SMST}$ = 3.8 µmol m$^{-2}$ s$^{-1}$and MBE: VPRM$_{GOSIF,SMST}$ = 3.2 µmol m$^{-2}$
s$^{-1}$, VPRM$_{TROPOSIF,SMST}$ = 2.4 µmol m$^{-2}$ s$^{-1}$) (see Table 6). The observed seasonal anomalies in
NEE ranges from -4.9 to 8. µmol m$^{-2}$ s$^{-1}$ with a standard deviation of 3.6 µmol m$^{-2}$ s$^{-1}$. These
variations are well captured by our model with a mean bias of 1.6 µmol m$^{-2}$ s$^{-1}$. The modifications
made to VPRM GPP and $R_{eco}$ fluxes improved the model's ability to capture NEE fluxes over Betul.
Since VPRM$_{TROPOSIF,SMST}$ is found to be closer to the observation among other modified
VPRM models, the rest of the analysis uses the simulations from VPRM$_{TROPOSIF,SMST}$ (hereafter
referred to as VPRM$_{refined}$).
**3.4 Flux spatial patterns**
We find strong spatial variations in the NEE and GPP estimates by VPRM$_{refined}$ over the
Indian region (see Figs. 6 and 7), with distinct zonal and meridional variations. These variations
are expected, resulting from factors such as patterns in annual mean temperature, precipitation, and
radiation which can have significant influences on the spatial pattern of ecosystem carbon fluxes
(e.g., Yu et al., 2013). The inter-annual variability in simulated NEE during the study period is
highly correlated ($R^2$>0.5) with the interannual variability in the country's precipitation pattern,
which is in line with Dadhwal, 2012. Annually, most of the country's biomes remained as a net
carbon sink, with higher NEE values over the southwest and northeast regions, which are





dominated by Evergreen and Mixed forest ecosystems. The highest GPP values are also found in
these regions, marking the highest productive biomes. Comparatively, high GPP is observed in the
eastern part of central India, where the Deciduous ecosystem prevails. However, respiration
exceeds primary productivity over the above region, leaving it as a carbon source on an annual
scale. A major part of the country shows moderate GPP values (~0.08 kg C m$^{-2}$ month$^{-1}$ to 0.15 kg
C m$^{-2}$ month$^{-1}$), while a large area is covered by cropland vegetation. Ecosystem productivity is
minimal in the northern and north western parts of the country under Shrubland vegetation.
During the period from 2012 to 2020, the Indian terrestrial biosphere acted as a net carbon
sink annually. The NEE value ranges from -0.38 Pg C yr$^{-1}$ to -0.51 Pg C yr$^{-1}$ in 2012 to -0.53 Pg C
yr$^{-1}$ to -0.64 Pg C yr$^{-1}$ in 2020 (see Fig. 6). The inter-comparison of total NEE fluxes are lower in
CT and TRENDY compared to those of VPRM$_{refined}$ ($\mu_{(VPRM_{refined}-CT)}$ = -0.34 Pg C yr$^{-1}$;
$\mu_{(VPRM_{refined}-CT)}$ = -0.25 Pg C yr$^{-1}$ in which $\mu$ represents sample mean of differences). The NEE
differences reported above used VPRM$_{refined}$ respiration model parameters calibrated using
FLUXNET. The corresponding NEE differences when using FLUXCOM are: $\mu_{(VPRM_{refined}-CT)}$ =
-0.52 Pg C yr$^{-1}$; $\mu_{(VPRM_{refined}-CT)}$ = -0.41 Pg C yr$^{-1}$. An ensemble means using 14 TRENDY
models is used for the analysis and the above reported values are based on the year 2018. Our NEE
estimates (see Fig. 6) are higher than the previously published studies in which process-based and
light-use efficiency models were used (Cervarich et al., 2016; Nayak et al., 2015; Rao et al., 2019).
Based on the CASA model, Nayak et al. (2015) estimated a NEE value of -0.0098 Pg C yr$^{-1}$ for a
26-year period from 1981 to 2006, showing ecosystem transition from a carbon source in the 1980s
to a carbon sink in the subsequent decades. Using a process-based model ensemble, namely
TRENDY, Cervarich et al. (2016) estimated an annual NEE value of -0.2 Pg C yr$^{-1}$ from 2000 to



2013. A similar study using TRENDY models by Rao et al. (2019) also showed the uptake capacity
of the Indian region by -0.14 Pg C yr$^{-1}$ from 1901 to 2010.

The spatial patterns for monthly averaged NEE and GPP are presented in Fig. 7. The highest

values for NEE and GPP are found during the months of July to September i.e., the summer
monsoon season and the lowest values are found during the dry and hot months from March to
May (Fig. 7). The Indo-Gangetic plain shows higher NEE values during the winter months and
summer monsoon seasons. This coincides with the highest productivity associated with the peak
growing stage of the two major cropping seasons in India. Enhanced NEE and GPP values across
the entire Indian region from June to September are associated with enhanced agricultural crop
production based on the availability of monsoonal rainfall. The south eastern part of the country
shows an increase in NEE and GPP values as a result of increased productivity upon the
commencement of the North East winter monsoon. Most parts of the country remained carbon
neutral from March to May. Winter crop harvesting and unfavourable conditions for plant growth
(e.g., high temperature, low water availability, low soil moisture content etc.) resulted in minimum
productivity during this period. A major part of Deciduous vegetation persisted as a source
throughout the seasons. Even though Deciduous vegetation shows higher seasonality and GPP
values, ecosystem respiration dominates GPP across this biome, leaving it as a carbon source or
carbon neutral on an annual scale (e.g., Deb Burman et al., 2021; Sarma et al., 2022).
**3.5 Derived ecosystem productivity and exchanges across different biomes**

Here, we present the derived ecosystem productivity and exchange fluxes across seven

vegetation classes used in VPRM (Table 7). Large variability in ecosystem productivity is found
on different temporal scales. On an annual scale, the Mixed forest vegetation shows the highest



(GPP = 6.35 kg C m$^{-2}$ yr$^{-1}$) productivity, followed by the Evergreen forest, Deciduous forest and
Savanna biomes (GPP = 5.51 kg C m$^{-2}$ yr$^{-1}$, 4.63 kg C m$^{-2}$ yr$^{-1}$, 4.60 kg C m$^{-2}$ yr$^{-1}$, respectively).
Figure 8 presents the spatial pattern in annually averaged GPP over different vegetation for the
year 2020. The GPP distribution is found to be spatially heterogeneous and is influenced by local
geographic and climatic factors. The spatial distribution of GPP also exhibits inter-annual
variations (see Fig. S2). As expected, lower productivity rates are found for Shrubland (1.74 kg C
m$^{-2}$ yr$^{-1}$) and Cropland (1.43 kg C m$^{-2}$ yr$^{-1}$). Cropland covers more than 68% of Indian land mass.
However, the total Cropland GPP is found to be lower than Deciduous forests (area coverage:
4.4%), Evergreen (area coverage: 4.8%) and Mixed forests (area coverage: 3.7%), while the total
area covered by these vegetation classes is small. The lowest annual productivity is seen over the
Grassland with a GPP value of 0.66 kg C m$^{-2}$ yr$^{-1}$. Even though higher productivity is associated
with Deciduous forest, this biome results in less net carbon uptake due to the high respiration fluxes
of this vegetation. The highest productivity of forest ecosystems over Grassland is also seen in
other parts of the globe (e.g., Yu et al., 2013). The contribution of each vegetation to the national
GPP budget also depends on the area covered by each vegetation. As a result, to the national GPP
budget of 3.88 Pg C yr$^{-1}$, for the year 2020, Cropland is the major contributor (49.6%), followed
by Evergreen forest (14.9%), Mixed forest (12.2%), Shrubland (12.0%), Deciduous forest (9.3%),
Savanna (1.1%) and Grassland (0.5%). Figure 9.a shows the annual mean GPP from different
vegetation classes from 2012 to 2020. On an annual scale, Mixed and Evergreen forest vegetation
groups show large GPP variability, while Cropland and Grassland exhibit lower GPP variability.
This variability across biomes remains consistent over the years during the analysis period.
Evergreen Forest is the largest contributor to the national NEE budget (~39.7%) followed
by Cropland (~33.6%), Mixed Forest (~31.5%), Shrubland (~10.7%), Savanna (~1.1%), Grassland



(~-0.2%), and Deciduous forest (~-17.3%), based on the data from 2020. The Evergreen forest and
Mixed forest vegetation are with the highest carbon fixation sink capacity, showing high NEE
values (see Table 7) (NEE of ~-2.4 kg C m$^{-2}$ yr$^{-1}$), followed by Savanna with an annual NEE value
of ~-1.3 kg C m$^{-2}$ yr$^{-1}$. A Moderate net carbon fixation efficiency (NEE of ~-0.3 and – 0.2 kg C m$^{-2}$
$^{2}$ yr$^{-1}$) is shown by Shrubland and Cropland vegetations, respectively. The above reported values
are based on VPRM$_{refined}$ in which respiration and model parameters are calibrated using
FLUXNET. The lowest efficiency is found for Deciduous vegetation, indicating a carbon-neutral
biome. Evergreen and Mixed forest ecosystems persisted as net sinks throughout the seasons with
higher productivity (Fig. 8). Figure 9.b shows the annual mean NEE from different vegetation
classes from 2012 to 2020. On an annual scale, Mixed and Evergreen forest vegetation groups
show large NEE variability while lowest by the Cropland and Grassland. Similar to GPP, the
variability found across biomes remains consistent over the years during the analysis period with
interannual variations. It is also seen that over the years sink capacity of most of the vegetation has
increased.
**3.6 Seasonal and diurnal cycles across different biomes**
Figure 10 shows the seasonal variations in VPRM$_{refined}$ simulated NEE fluxes across
different biomes from 2012 to 2020. The seasonality varies across the vegetation. Vegetations such
as Cropland, Savanna, and Shrubland show similar seasonal carbon dynamics with higher NEE
from September to October and lower NEE from April to May. These biomes remained as carbon
sinks throughout the year except for March to May. On the other hand, Grassland shows higher
NEE from July to August and lower NEE from November to January. Even though Mixed forests
show seasonal variations, it is not consistent over the years. Throughout the year, Grassland,
Cropland, Evergreen forest, and Mixed forest remained as a net carbon sink. On the other hand,
Deciduous vegetation remained a carbon source as ecosystem respiration surpassed primary



production. Strong seasonality in NEE is exhibited by Savanna (11.94 µmol m$^{-2}$ s$^{-1}$), followed by
Mixed forest (10.57 µmol m$^{-2}$ s$^{-1}$), while the least is observed for Cropland (3.38 µmol m$^{-2}$ s$^{-1}$)
(Statistics presented for the year 2020). For each vegetation, the spatial heterogeneity in NEE
values is more during those months showing higher uptake capacity (Fig. not shown).

We find that the seasonality in the ecosystem uptake is associated with the wet and dry

conditions, showing a transition from dry and cooler winter months to wet and hot summer months
(see Fig. S3). The majority of the vegetation shows higher productivity during August to September
and lowest during March to May (e.g., Cropland, Savanna, Deciduous forest, Evergreen forest,
Mixed forest, and Shrubland). The ecosystems show a semi-annual cycle with a primary
productivity peak during the winter months (December - January) and a secondary peak during the
monsoon season (August - September). Productivity of Grassland remained high from June
onwards and lasted till August. For 2020, the Savanna shows strong seasonality with 18.6 µmol m$^{-2}$ s$^{-1}$
variation in GPP value from low to high productive month followed by Deciduous and Mixed
forest groups (16.57 µmol m$^{-2}$ s$^{-1}$, 12.01 µmol m$^{-2}$ s$^{-1}$, respectively). Grassland shows the lowest
variation in GPP with the season (3.83 µmol m$^{-2}$ s$^{-1}$). Also, the magnitude of seasonal variability
remains low for vegetations such as Cropland (5.05 µmol m$^{-2}$ s$^{-1}$) and Savanna (5.29 µmol m$^{-2}$ s$^{-1}$

).

Figure 11 shows the diurnal variations in VPRM$_{refined}$ simulated GPP fluxes at a monthly

scale for different vegetation classes during 2020. The diurnal variability of GPP varies with the
season. A seasonal shift in the peak uptake time is found, and it varies with vegetation. Larger
productivity is found during noon hours (10:00 -14:00 local time), of summer monsoon months of
August and September and the post-monsoon months of October and November. The productivity
gradually decreases with the progress of the dry season. The lowest GPP values are found during
March and May. Strong daytime variability, with peak uptake during early morning hours and weak



uptake during afternoon hours, is also found during this dry season, indicating the temperature
dependence on ecosystem productivity, which also varies with biome type and age.

**4. Conclusion:**

This study presents the terrestrial flux distribution of $CO_2$ over India on a 0.1°×0.1° grid at
a temporal resolution of 1 hour from 2012 to 2020. We utilise satellite-based vegetation and
ecosystem productivity indices and high-resolution meteorological data in a data-driven biospheric
model to improve the model estimates of terrestrial biosphere $CO_2$ flux components over India. In
particular, we take advantage of satellite missions, such as TROPOMI and OCO-2 providing
retrievals of solar-induced chlorophyll fluorescence (SIF) and relate them to ecosystem
productivity across different biomes. The derived flux products better explain the magnitude and
fine-scale variability over the region compared to other existing model estimates.
We investigated how our model captures the seasonal pattern in NEE and GPP compared
to other biospheric models with different model structures, such as the inversion product CT and
the ensemble of process-based models TRENDY. Though VPRM$_{STD}$ shows better agreement with
observations in predicting the seasonality of NEE fluxes ($R^2 = 0.59$) than CT ($R^2 = 0.24$) and
TRENDY ($R^2 = 0.45$) for the period from 2012 to 2018, the simulations considerably
underestimated the NEE fluxes at a monthly scale, with model biases of 3.2 µmol m$^{-2}$ s$^{-1}$ for NEE
and -6.7 µmol m$^{-2}$ s$^{-1}$ for GPP. The model-observation bias is high for simulating GPP during
productive months (June - December). We infer that the GPP underestimation by VPRM$_{STD}$ can
be related to the MODIS reflectance products and the plausible errors in model parameters. The
VPRM$_{STD}$ model parameters are not optimised using flux tower measurements due to the
unavailability of flux observations over the Indian sub-continent, thereby limiting the model
performance over the domain while using uncalibrated model.




We performed biome-specific analyses of SIF products, deducing their spatial and temporal

characteristics over Indian biomes and applied them to VPRM$_{STD}$. Compared to other process-
based biospheric models and atmospheric inversion products, the refined VPRM shows remarkable
performance in explaining small-scale variability. By improving GPP and R$_{eco}$ simulations, the
model has improved its ability to capture the observed NEE fluxes (R$^2$>0.5) with a significant
reduction in RMSE (~3 µmol m$^{-2}$ s$^{-1}$) and MBE (~3 µmol m$^{-2}$ s$^{-1}$) values. While evaluating
VPRM$_{refined}$ GPP with observation-based GPP at the Betul site, we find better model performance
compared to VPRM$_{STD}$ with reduced bias (RMSE = 4.3 µmol m$^{-2}$ s$^{-1}$ and MBE = -2.6 µmol m$^{-2}$ s$^{-1}$
$^{1}$). The monthly variations in GPP (R$^2$>0.7) and R$_{eco}$ (R$^2$>0.8) are better captured by VPRM$_{refined}$
than other models. The VPRM$_{refined}$ reproduces the seasonal anomalies exhibited by Betul
observations remarkably well, for example, with explained variability of GPP and NEE anomalies
by 85% and 68%, respectively from 2014 to 2018. However, the model evaluation is limited only
to the Deciduous ecosystem due to the observational constraints that are only representative of the
above ecosystem.

We find significant spatial variations in the NEE and GPP flux distributions simulated by

VPRM$_{refined}$, which are associated with the spatial heterogeneity in annual mean temperature,
precipitation, and radiation. Evergreen and Mixed forests covering southwest and northeast of India
show the highest productivity annually. Ecosystem productivity is minimal in the northern and
north western parts of the country (mainly Shrubland vegetation). The Deciduous forest remained
as an annual carbon source despite the high productivity due to higher respiratory fluxes. NEE and
GPP fluxes show higher values during July to September (i.e., the summer monsoon season) and
lower values during March to May (dry and hot months), and these seasonal variations are in line
with the seasonal variations in the rain, temperature, and solar radiation. Since more than 60% of
the country is covered with Croplands, the agricultural pattern also influences the seasonality in





GPP and NEE. Overall, we find that the Indian biosphere acts as a sink with an annual NEE ranging
from -0.38 Pg C yr$^{-1}$ (-0.51 Pg C yr$^{-1}$) to -0.53 Pg C yr$^{-1}$ (-0.88 Pg C yr$^{-1}$) when the respiration
model parameters calibrated using FLUXNET (FLUXCOM) and an annual GPP ranging 3.39 yr$^{-1}$
to 3.88 Pg C yr$^{-1}$ for the years from 2012 to 2020.

Though we have demonstrated the use of additional satellite-based observations and

provided the high-resolution gridded $CO_2$ flux distributions, future work evaluating the simulated
flux distribution with an adequate number of flux site observations and atmospheric $CO_2$ mixing
ratio is warranted. Potential improvements to VPRM include i) further refinement in the ecosystem
respiration accounting for moisture and heat stress and other biomass disturbance and ii)
incorporating flux observations from different ecosystems to enhance the flux representativeness
with better empirically derived and biome-specific model parameters. The increased number of
flux tower observations in the future will help to optimise the model parameters to enhance the
robustness of these simulations.

Given the considerable difference in flux components among the terrestrial biospheric

models, the analyses demonstrated here can guide future model improvements in deriving GPP and
ecosystem respiration. By showing the potential of VPRM model to predict the observed variations
in GPP better than solely SIF-based GPP products, the present study demonstrates the way to
calibrate the VPRM model parameters in the absence of eddy covariance measurements. The next
step would be to combine atmospheric data and VPRM through inverse modelling to better
understand the Indian carbon balance.






**Data availability**

The VPRM simulations will be made available upon request to the corresponding author. The Carbon Tracker (CT2019B) is freely available online at https://gml.noaa.gov/ccgg/carbontracker/CT2019B/. TRENDYv10 datasets used in this study are available upon request to S. Sitch. Eddy covariance observation data may be available upon request to NRSC; https://www.nrsc.gov.in/. The TROPOMI data is available online at http://ftp.sron.nl/open-access-data-2/TROPOMI/tropomi/sif/v2.1/l2b/. GOSIF_v2 datasets used are available freely from http://data.globalecology.unh.edu/. ERA5 data used is freely available at https://cds.climate.copernicus.eu/cdsapp#!/dataset/reanalysis-era5-land?tab=overview. GLEAM v3 data is available freely at https://www.gleam.eu/#datasets. FLUXNET data is available freely from https://db.cger.nies.go.jp/DL/10.17595/20200227.001.html.en. FLUXCCOM data used is freely available from https://www.bgc-jena.mpg.de/geodb/projects/DataDnld.php. TRMM precipitation data used is available freely from https://disc.gsfc.nasa.gov/datasets/TRMM_3B42_Daily_7/summary.

**Authors contribution**

Aparnna Ravi: Method development, Coding, Data processing, Analysis, Visualization, Writing – original draft preparation, Dhanyalekshmi Pillai: Conceptualization, Method development, and Writing - review & editing, Christoph Gerbig: Data processing and Writing - review & editing, Vishnu Thilakan: Analysis and Writing - review & editing, Stephan Sitch: Model data and Writing - review & editing, Sönke Zaehle: Writing - review & editing, Chandrashekhar Jha: EC flux tower data acquisition and processing and Writing - review & editing

**Declaration of Competing Interest**

The authors affirm that they have no known financial or interpersonal conflicts that would have appeared to have an impact on the research presented in this study.



**Acknowledgements**
This study has been funded by the Max Planck Society allocated to the Max Planck Partner Group
at IISERB. DP acknowledges the support from the Science and Engineering Research Board
(SERB) through an Early Career Research Award (grant no. ECR/2018/001111) for generating
some data products used in the study. AR acknowledges the support of IISERB's high-performance
cluster system for computations, data analysis, and visualization. AR and VT are grateful to the
Ministry of Human Resource Development (MHRD, India) for their PhD scholarships. We thank
National Remote Sensing Centre (NRSC), Hyderabad, for providing access to Betul EC flux tower
data, and we acknowledge the efforts of scientists and technicians from the Forestry and Ecology
Group at NRSC Hyderabad for the EC data acquisition.





**Tables**
**Table 1: An overview of the observational- and model- based datasets used in this study.**

| Dataset | Products | Spatial resolution | Temporal resolution | Period | Reference |
|---|---|---|---|---|---|
| VPRM | GPP, NEE, $R_{eco}$ | $0.1° \times 0.1°$ | Hourly | 2012 - 2020 | (Mahadevan et al., 2008) |
| TROPOMI | SIF | $0.1° \times 0.1°$ | Hourly | 2018 - 2020 | (Köhler et al., 2018) |
| GOSIF_v2 | SIF | $0.05° \times 0.05°$ | 8 day | 2016-2020 | (Li & Xiao, 2019a) |
| ERA5 | ST | $0.1° \times 0.1°$ | Hourly | 2012-2020 | (Hersbach et al., 2020) |
| GLEAM v3 | SM | $0.25° \times 0.25°$ | Daily | 2012-2020 | (Martens et al., 2017) |
| HRLDAS | ST and SM | $0.03^0 \times 0.03^0$ | 3 hourly | 2012- 2017 | (Chen et al., 2007) |
| Gridded $R_{eco}$ from FLUXNET | $R_{eco}$ | $0.1^0 \times 0.1^0$ | 10 days | 2012-2019 | (Zeng, Jiye, 2020) |
| FLUXCOM | $R_{eco}$ | $0.5^0 \times 0.5^0$ | Monthly | 2012-2019 | (Jung et al., |



| | | | | | 2020) |
| --- | --- | --- | --- | --- | --- |
| EC | NEE, GPP, $R_{eco}$ | 1 km$^2$ | Half hourly | 2012-July 2019 | (Jha et al., 2013) |
| CT2019B | NEE | 1°✕1° | Three hourly | 2012 – March 2019 | (Peters et al., 2007) |
| TRENDYv 10 | NEE, GPP, $R_{eco}$ | Vary with model | Monthly | 2012 - 2020 | Ref. Table 2 |
| GOSIF_GP P_v2 | GPP | 0.05°✕0.05° | 8 day | 2016-2020 | (Li & Xiao, 2019b) |
| TRMM | Rainfall | 0.25°✕0.25° | Daily | 2016 - 2019 | (Kummerow et al., 2000) |













**Table 2. List of VPRM (both standard and refined) parameters and vegetation classes used**
**in this study. a. respiration model parameters calibrated FLUXNET; b. respiration model**
**parameters calibrated using FLUXCOM.***

| Vegetation class | $\lambda$ | $\alpha$ | $\beta$ | $SW_{down0}$ | $\eta_{vg}$ | $T_{s,vg}$ | | $M_{s,vg}$ | | $R_{vg}$ | |
|---|---|---|---|---|---|---|---|---|---|---|---|
| | | | | | | $a_T$ | $b_T$ | $a_M$ | $b_M$ | $a_R$ | $b_R$ |
| Grassland | 0.1334 | 0.0269 | 0 | 157 | 3.2945 | -0.0023 | 0.0004 | 2790.4 | 1320.2 | 3.96 | 2.9 |
| Cropland | 0.1209 | 0.0043 | 0 | 646 | 1.6002 | -0.0008 | -0.001 | 8588.3 | 7835.9 | 0.20 | 0.094 |
| Savanna | 0.1141 | 0.0049 | 0 | 682 | 3.7301 | -0.0009 | -0.003 | 10321.2 | 9546.6 | -0.07 | -0.01 |
| Shrubland | 0.0874 | 0.0239 | 0 | 303 | 3.3241 | -0.001 | 0.002 | 5059.4 | 2749.0 | 0.72 | 0.4 |
| Deciduous forest | 0.2555 | 0.3422 | 0 | 206 | 2.4613 | -0.043 | -0.043 | 29429 | 29429 | 2.502 | 2.502 |
| Evergreen forest | 0.1729 | 0.3258 | 0 | 324 | 1.788 | 0.005 | -0.003 | 4505.6 | 6906.2 | 0.44 | 0.4 |
| Mixed Forest | 0.2101 | 0.1601 | 0 | 501 | 2.3238 | -0.005 | -0.01 | 10214.6 | 10469.6 | 0.31 | 0.3 |

*Units are as follows: $\lambda$ :$\mu$mol $CO_2$ m$^{-2}$ s$^{-1}$/ $\mu$mol $SW_{down}$ m$^{-2}$ s$^{-1}$; $\alpha$: $\mu$mol $CO_2$ m$^{-2}$ s$^{-1}$/°C; $\beta$: $\mu$mol
$CO_2$ m$^{-2}$ s$^{-1}$; $SW_{down0}$: $\mu$mol m$^{-2}$ s$^{-1}$; $T_{s,vg}$: $\mu$mol $CO_2$ m$^{-2}$ s$^{-1}$ K$^{-1}$; $M_{s,vg}$: $\mu$mol $CO_2$ m$^{-2}$ s$^{-1}$ m$^{-3}$ m$^3$;
$\eta_{vg}$ and $R_{vg}$: dimensionless.




**Table 3: Spatial and temporal resolutions of the 14 dynamic global vegetation models from**
**TRENDY. The annual NEE and GPP fluxes of India from individual models, calculated as**
**the cumulative sum of corresponding fluxes at the models' original resolution in Pg C yr[1] are**
**also given.**

| Model | Spatial resolution | Temporal resolution | Reference | NEE (Pg C yr[-1]) | GPP (Pg C yr[-1]) |
|---|---|---|---|---|---|
| ISBA-CTRIP | 1°✕1° | Monthly | (Decharme et al., 2019) | -0.47 | 3.7 |
| SDVGM | 0.5°✕0.5° | Monthly | (Woodward et al., 1995) | -0.14 | 2.7 |
| IBIS | 1°✕1° | Monthly | (Foley et al., 2003; Kucharik et al., 2000) | -0.05 | 2.9 |
| VISIT | 0.5°✕0.5° | Monthly | (Kato et al., 2013) | -0.21 | 2.9 |
| CABLE-POP | 1°✕1° | Monthly | (Haverd et al., 2013) | -0.007 | 2.7 |
| ORCHIDEEv | 0.5°✕0.5° | Monthly | (Lurton et al., | -0.34 | 3.1 |





| 3 | | | 2020) | | |
|---|---|---|---|---|---|
| CLM5.0 | 1.25°✕0.942° | Monthly | (Buzan et al., 2015) | -0.24 | 2.1 |
| DLEM | 0.5°✕0.5° | Monthly | (Tian et al., 2015) | -0.45 | 3.5 |
| ISAM | 0.5°✕0.5° | Monthly | (Meiyappan et al., 2015) | -0.06 | 2.2 |
| JSBACH | 1.875°✕1.875° | Monthly | (Goll et al., 2015); (Reick et al., 2013) | -0.21 | 4.5 |
| LPX-Bern | 0.5°✕0.5° | Monthly | (Spahni et al., 2013; Stocker et al., 2013) | -0.07 | 2.9 |
| OCN | 1°✕1° | Monthly | (Zaehle & Friend, 2010) | -0.12 | 3.5 |
| ORCHIDEE | 0.5°✕0.5° | Monthly | (Krinner et al., 2005) | -0.32 | 2.6 |
| LPJ | 0.5°✕0.5° | Monthly | (Sitch et al., 2003) | -0.05 | 2.6 |



**Table 4: An overview of the eddy flux tower site, Betul.**

| Site Name | Sukhwan, Betul |
|---|---|
| Country | India |
| State | Madhya Pradesh |
| Location | 21°51'46.84" N, 77°25'33.67" E |
| Area | 1.76 km$^2$ |
| Vegetation type | Deciduous forest |
| Canopy height | 22 m |
| Tower height | 34 m |
| Annual precipitation | 1016 mm |
| Mean air temperature | 27 °C |
| Dominant species | Tectona grandis, Miliusa tomentosa |









**Table 5: Biome-specific scalars used for the conversion of TROPOSIF to GPP$_{TROPOSIF}$ across**
**different vegetation classes (see Sect. 2.2).**

| Vegetation | $\gamma_{TROPOSIF,vg}$ (mW m$^{-2}$ sr$^{-1}$ nm$^{-1}$)/ (µmol m$^{-2}$ s$^{-1}$) | $C_{TROPOSIF,vg}$ |
|---|---|---|
| Grassland | 7.84 | 0.40 |
| Cropland | 4.81 | 0.22 |
| Savanna | 5.12 | 0.32 |
| Shrubland | 5.00 | 0.39 |
| Deciduous forest | 5.35 | 0.34 |
| Evergreen forest | 5.47 | 0.64 |
| Mixed forest | 5.59 | 0.61 |










**Table 6: Comparison of monthly averaged NEE, GPP and R$_{eco}$ fluxes from VPRM model**
**simulations against EC observations for Betul from 2012 to 2018. Also reporting values for**
**2018, the only common year for which the SIF, and EC data are available.**

| Model vs Observations | 2012 - 2018 ($\mu$mol m$^{-2}$ s$^{-1}$) | | | 2018 | | |
|---|---|---|---|---|---|---|
| | $R^2$ | RMSE | MBE | $R^2$ | RMSE | MBE |
| GPP | | | | | | |
| VPRM$_{STD}$ | 0.71 | 8.3 | -6.7 | 0.74 | 8 | 6.2 |
| VPRM$_{GOSIF}$ | 0.71 | 4.9 | -3.4 | 0.74 | 4.1 | 2.57 |
| VPRM$_{TROPOSIF}$ | 0.71 | 4.3 | -2.6 | 0.74 | 3.6 | 1.77 |
| R$_{eco}$ | | | | | | |
| VPRM$_{STD}$ | 0.02 | 5.7 | -3.5 | 0.01 | 4.9 | -2.9 |
| VPRM$_{ST}$ | 0.06 | 4.4 | 0.08 | 0.16 | 3.8 | 0.7 |





| | | | | | | |
|---|---|---|---|---|---|---|
| VPRM$_{SM}$ | 0.80 | 2.0 | -0.01 | 0.84 | 1.6 | 0.4 |
| VPRM$_{MOD(STSM)}$ | 0.82 | 1.9 | -0.01 | 0.88 | 1.4 | 0.2 |
| NEE | | | | | | |
| VPRM$_{STD}$ (-GPP $_{(VPRMSTD)}$ + R$_{eco(VPRMSTD)}$) | 0.59 | 4.4 | 3.2 | 0.65 | 5.2 | 3.3 |
| VPRM$_{GOSIF,SMST}$ (-GPP$_{(VPRMGOSIF)}$+ R$_{eco(VPRMMOD(STSM))}$) | 0.53 | 4.4 | 3.2 | 0.66 | 4.3 | 2.8 |
| VPRM$_{TROPOSIF,SMST}$ (-GPP$_{(VPRMTROPOSIF)}$+ R$_{eco(VPRMMOD(STSM))}$) | 0.56 | 3.8 | 2.4 | 0.68 | 3.7 | 2 |
| TRENDY | 0.45 | 3.3 | 1.1 | 0.51 | 3.6 | 1.4 |
| CT | 0.24 | 3.5 | 1.2 | 0.17 | 4 | 1.4 |






**Table 7. Biome specific annual fluxes from VPRM_refined in kg C m$^{-2}$ yr$^{-1}$ and total fluxes in Pg**
**C yr$^{-1}$ are provided for the year 2020. The reported NEE values used respiration model**
**parameters calibrated using FLUXNET.**

| | | | | | | Evergreen | Mixed | Deciduous |
|---|---|---|---|---|---|---|---|---|
| | | Grassland | Cropland | Savanna | Shrubland | Forest | Forest | Forest |
| | kg C m$^{-2}$ yr$^{-1}$ | 0.11 | -0.28 | -1.31 | -0.39 | -2.42 | -2.65 | 2.70 |
| NEE | Pg C yr$^{-1}$ | 0.005 | -0.30 | -0.01 | -0.07 | -0.31 | -0.22 | 0.31 |
| | kg C m$^{-2}$ yr$^{-1}$ | 0.66 | 1.43 | 4.60 | 1.74 | 5.51 | 6.35 | 4.63 |
| GPP | Pg C yr$^{-1}$ | 0.03 | 2.60 | 0.062 | 0.63 | 0.78 | 0.64 | 0.49 |
| | kg C m$^{-2}$ yr$^{-1}$ | 0.69 | 1.19 | 2.92 | 1.35 | 3.05 | 3.4 | 5.71 |
| R$_{eco}$ | Pg C Yr$^{-1}$ | 0.03 | 2.21 | 0.04 | 0.51 | 0.36 | 0.32 | 0.66 |




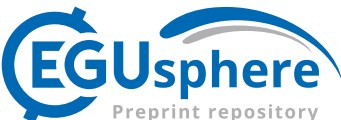

**Figures**

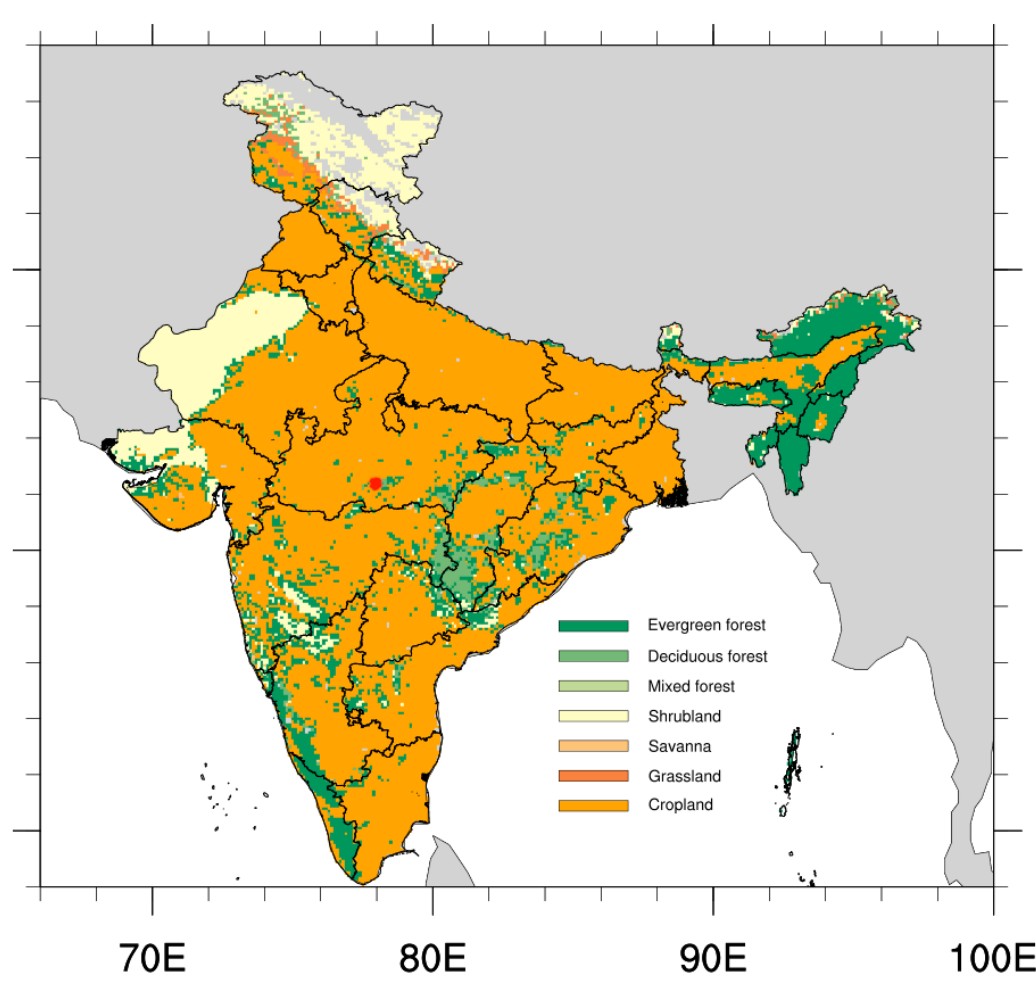


**Fig. 1: An overview of the major vegetation classes for the study region. Solid red circle**
**denotes the Eddy covariance observation site at Betul.**





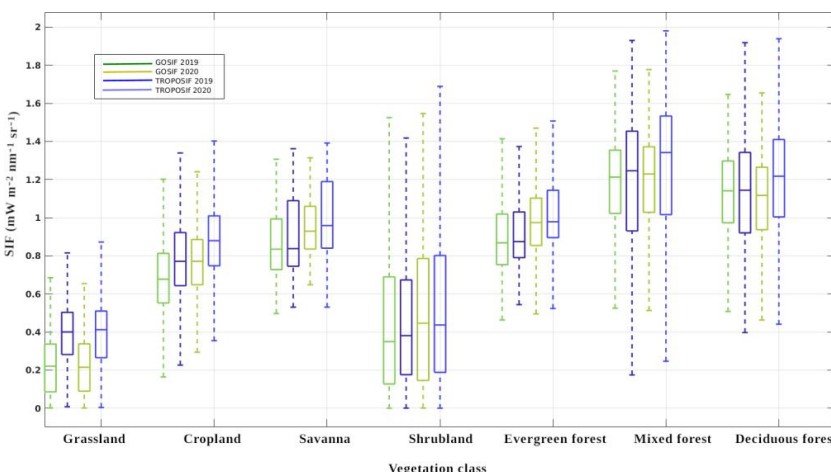

**Fig. 2: Comparison between annually averaged SIF retrievals from OCO-2 (GOSIF) and TROPOSIF based products across vegetation classes over India for 2019. GOSIF (estimated at 757 nm) are scaled by respective biome-specific scaling factors (see Table. 5) to compare with TROPOMI SIF (estimated at 757 nm and 771 nm).**



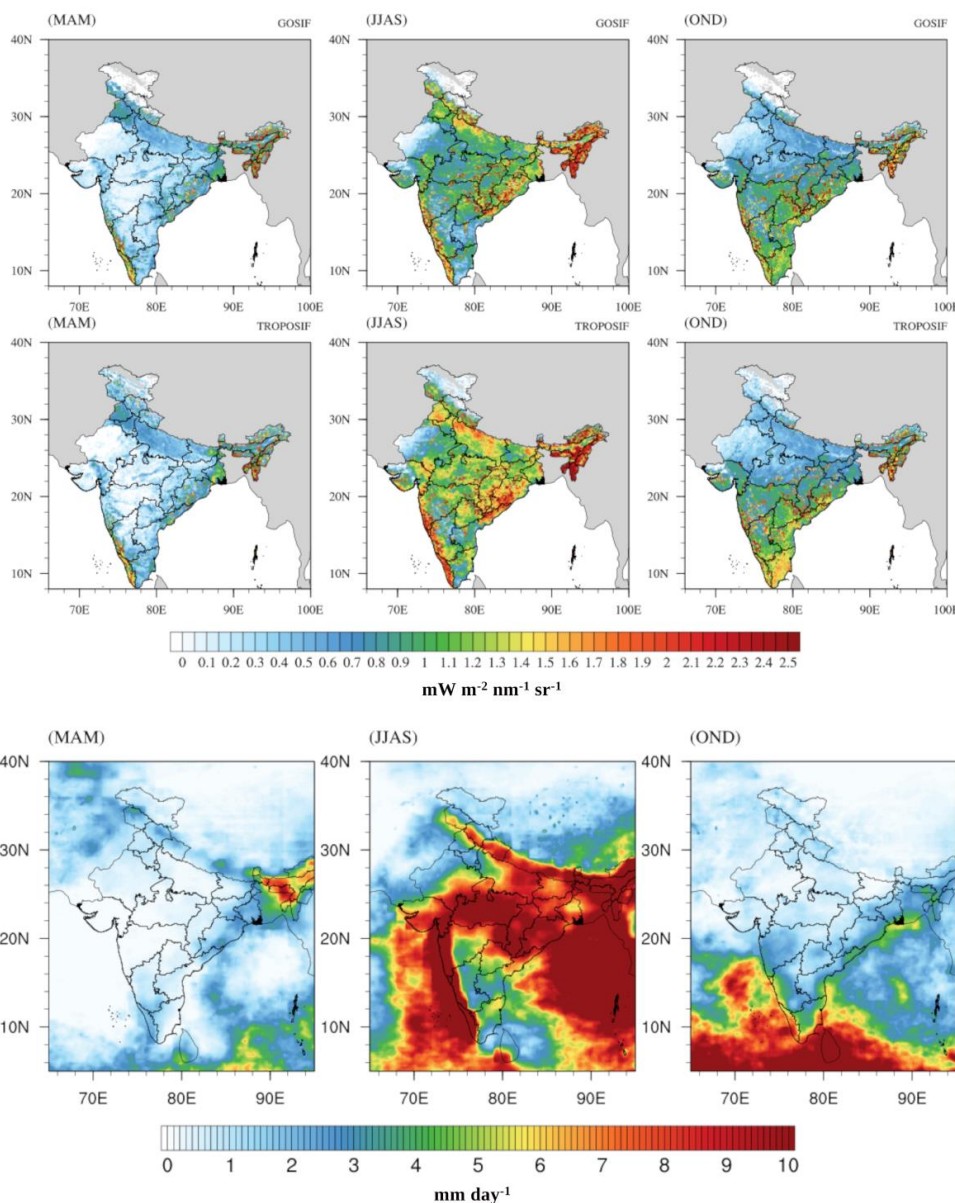


**Fig. 3: Seasonal distribution patterns of SIF and precipitation over India for the year 2019:**

**First row: GOSIF, Second row: TROPOSIF, and Third row: TRMM precipitation data,**

**respectively.**



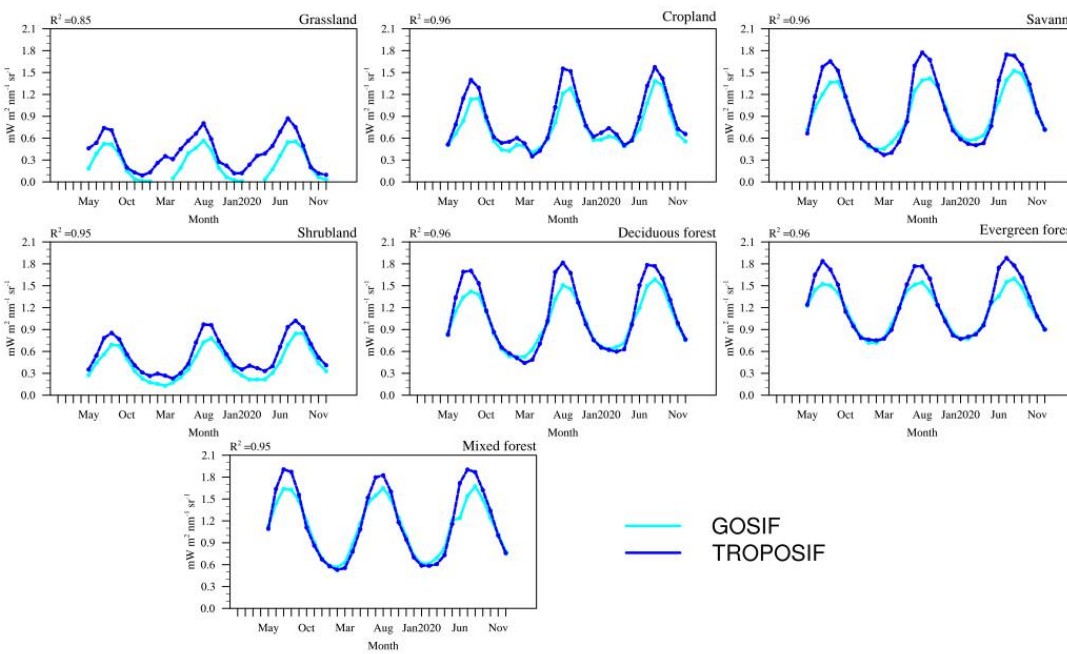


**Fig. 4: Time series of monthly averaged SIF (GOSIF and TROPOSIF) across different**

**biomes over India from 2018 to 2020. The vegetation classification based on SYNMAP is**

**used to represent SIF for different biomes.**

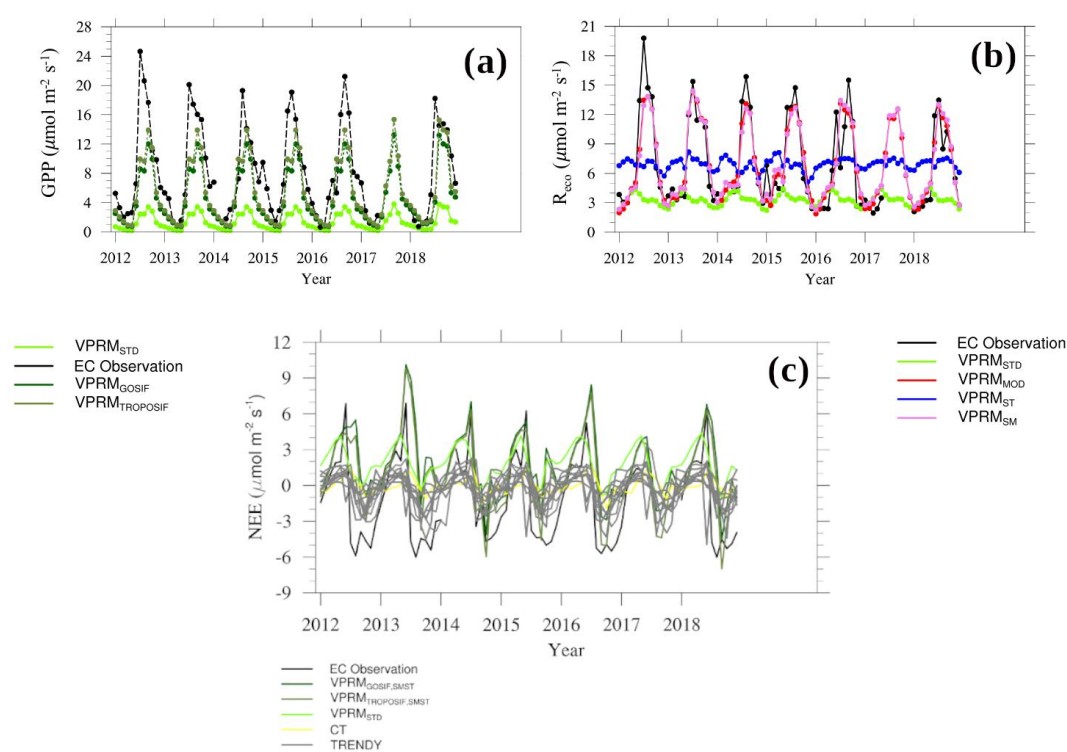


**Fig. 5: Comparison of monthly averaged EC observations with a) GPP, b) R$_{eco}$, and c) NEE**

**simulations over Betul for the period 2012 to 2018.**



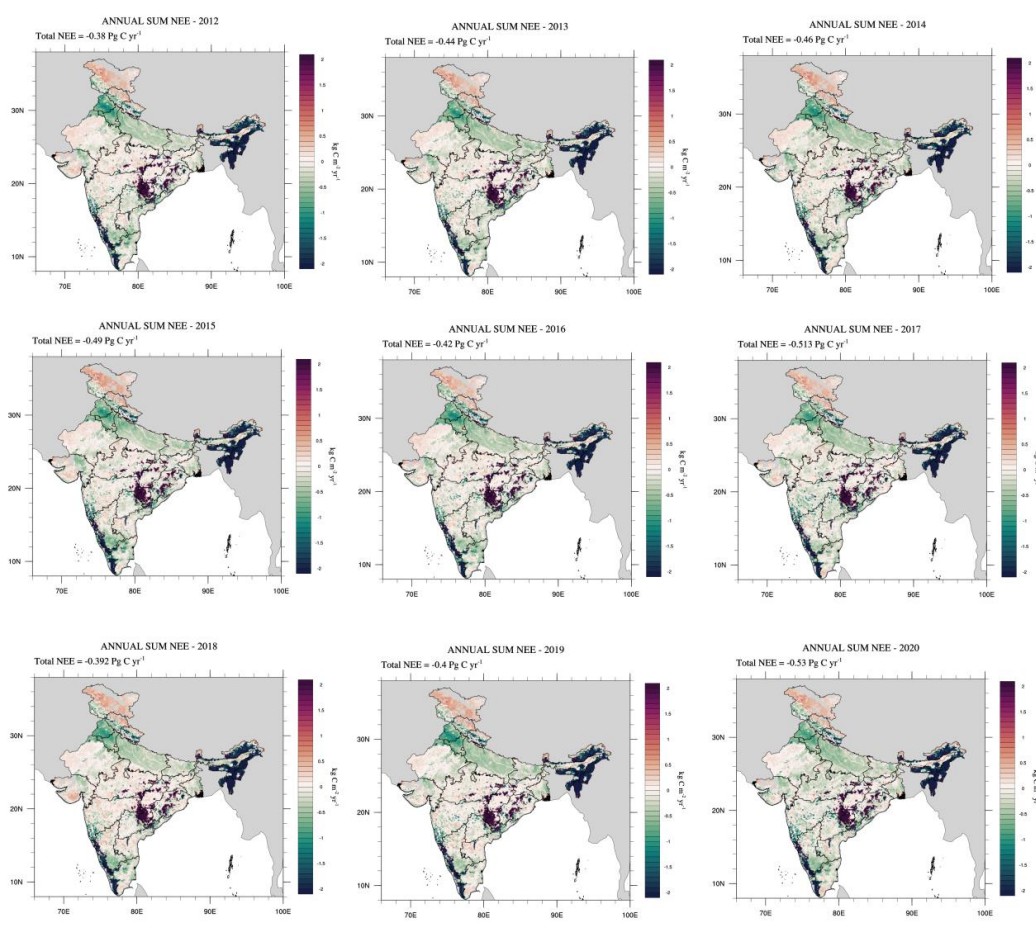

**Fig. 6: Spatial patterns in annual NEE fluxes as simulated by VPRM$_{refined}$ over the Indian region for the years from 2012 to 2020. The shown NEE values used respiration model parameters calibrated using FLUXNET.**



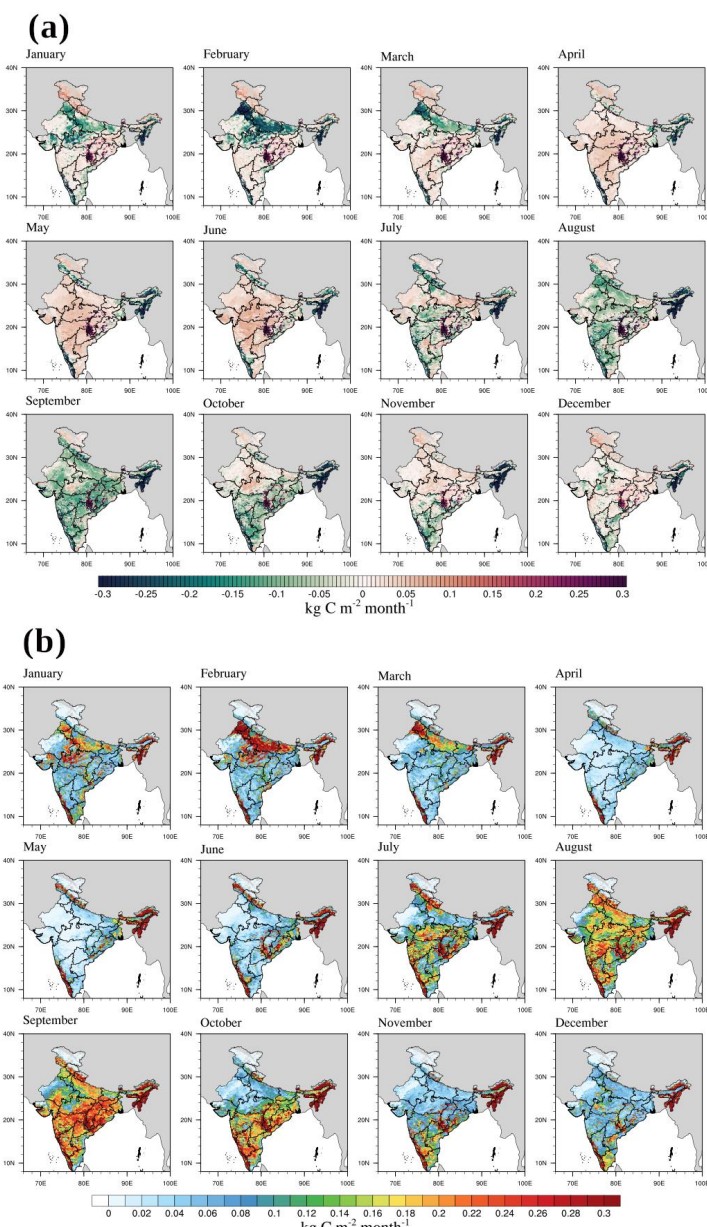

798

**Fig. 7: Spatial pattern in monthly averaged fluxes from VPRM$_{refined}$ for the year 2020. a)**

**NEE and b) GPP. The shown NEE values used respiration model parameters calibrated**

**using FLUXNET.**



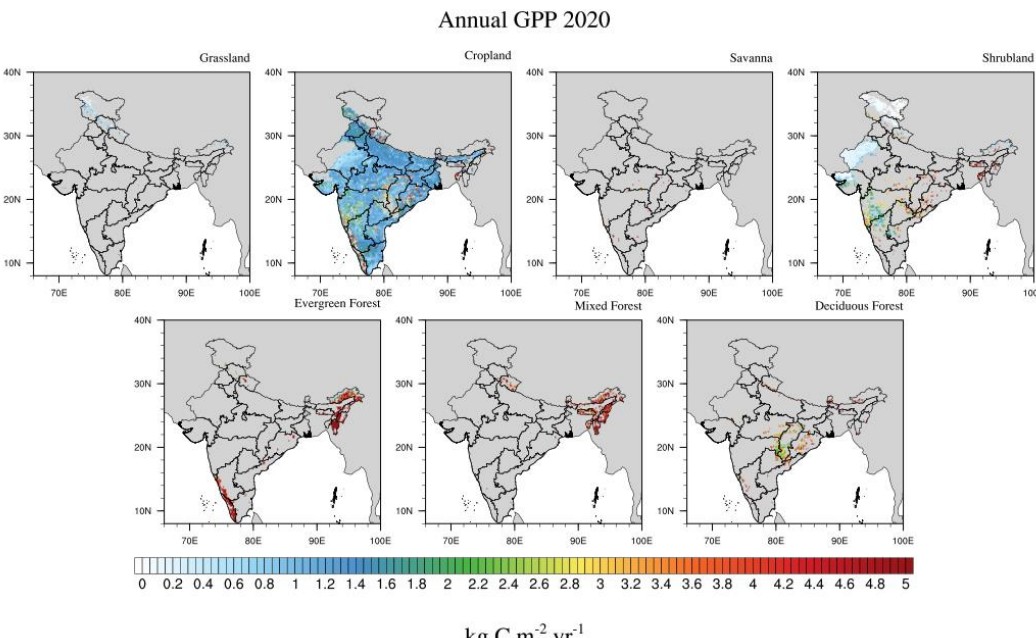

802

**Fig. 8: Spatial pattern in the annual GPP from VPRM$_{refined}$ over different vegetation for the**

**year 2020.**





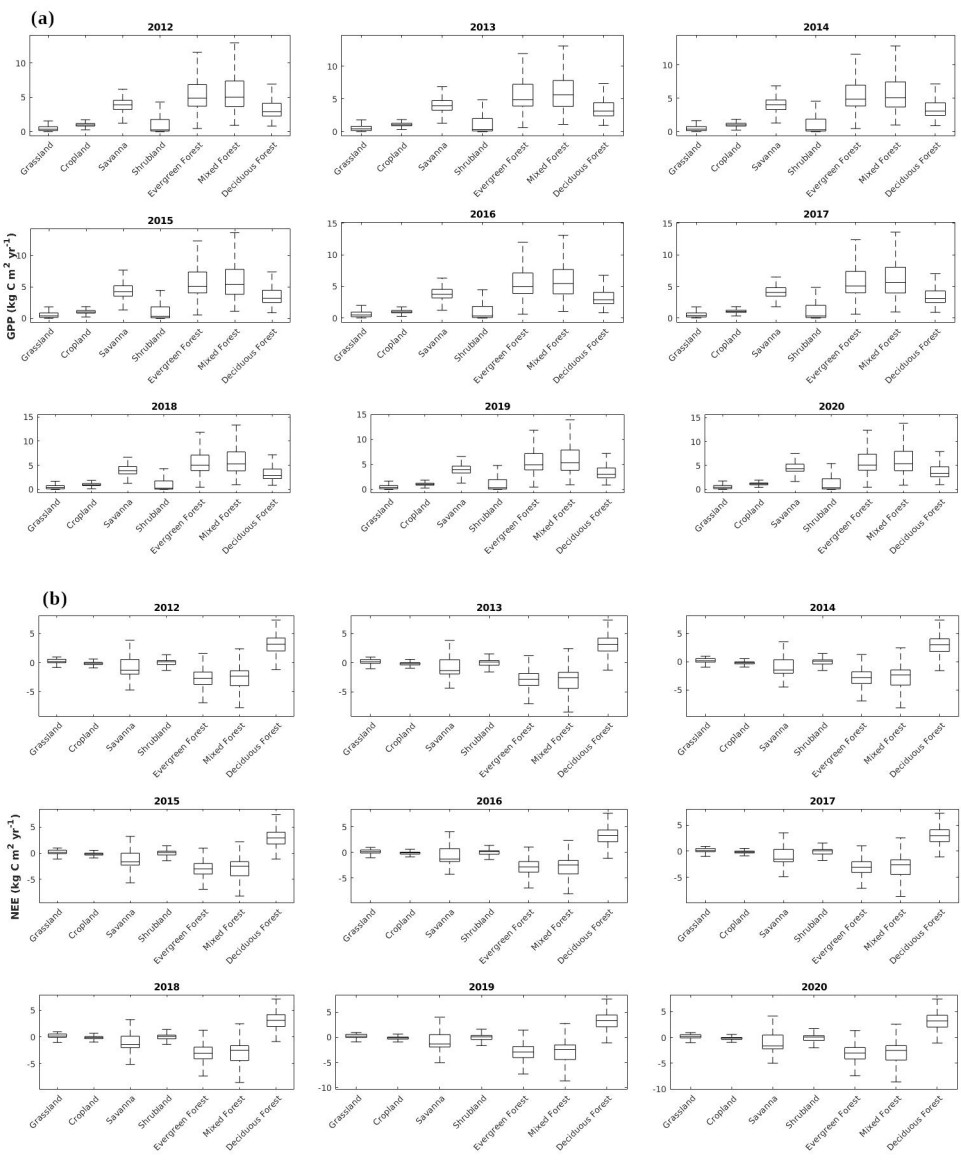

805

**Fig. 9: The biome-specific annual VPRM$_{refined}$ a) GPP and b) NEE from 2012 to 2020.**

**Upper and lower limit of the box shows 25$^{th}$ and 75$^{th}$ percentile of the data and center line**

**shows the median. All the values which are 1.5 times higher than the 25th and 75th**

**percentile are considered as outliers and are removed from the graph. The shown NEE**

**values used respiration model parameters calibrated using FLUXNET.**

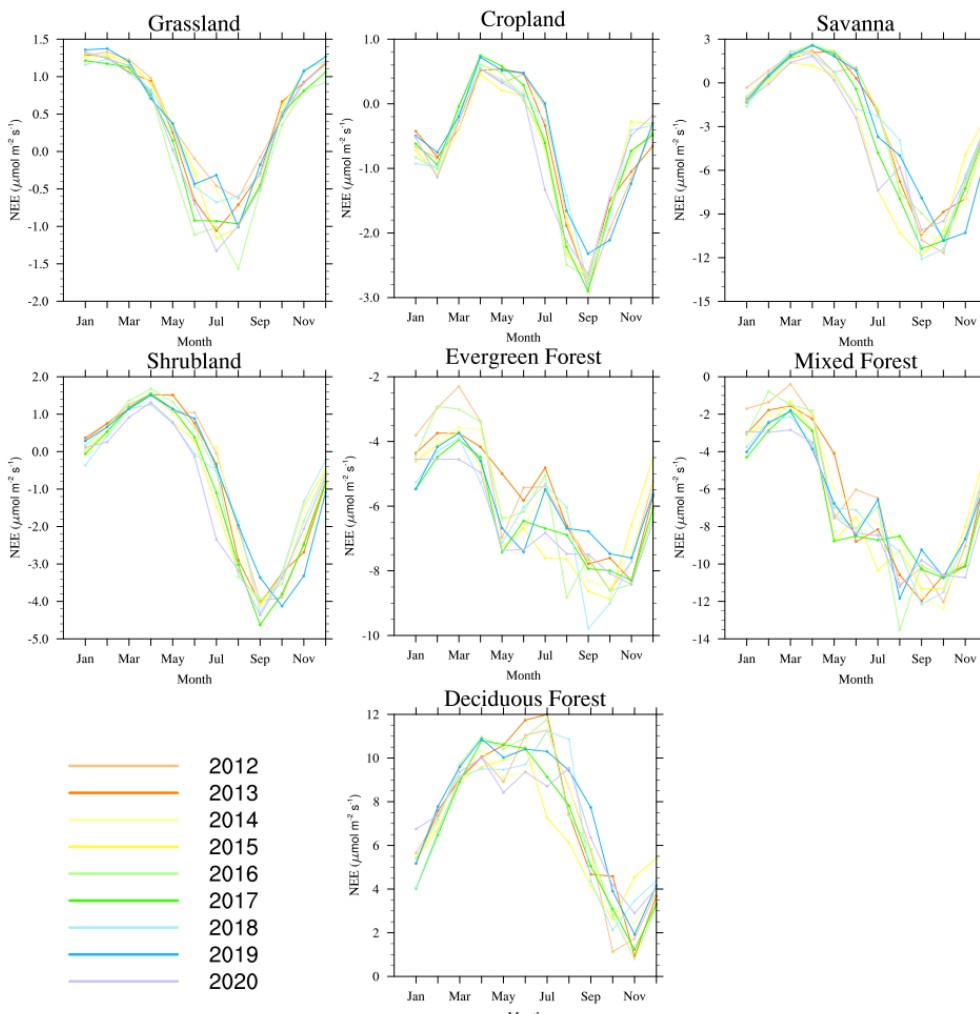


**Fig. 10: Temporal variations in monthly averaged NEE fluxes from VPRM$_{refined}$ for the**

**years 2012 to 2020. The shown NEE values used respiration model parameters calibrated**

**using FLUXNET.**




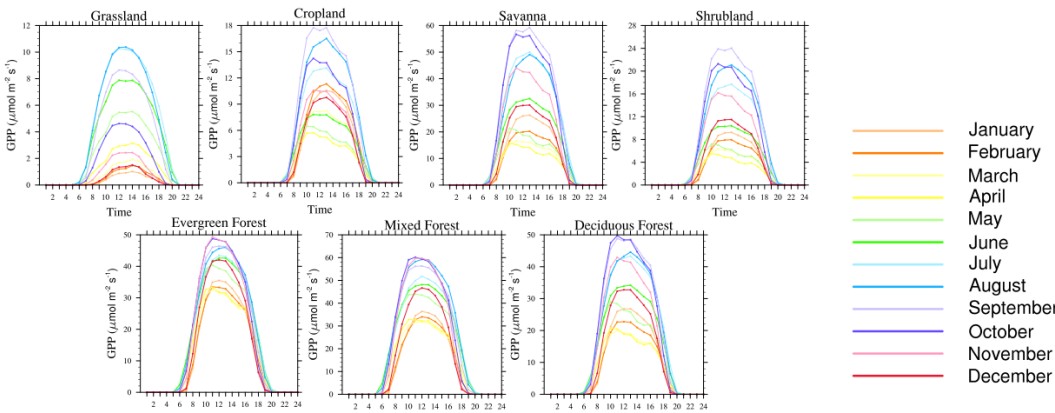


**Fig. 11: Diurnal variations in VPRM$_{refined}$ GPP fluxes during 2020.**













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
