# Peer review of "Spatiotemporal variations in terrestrial biospheric CO2 fluxes of India derived from"

_EGUsphere, 2023_

## Author Comment (AC2)

**Response to Reviewers' Comments**

We are greatly thankful to Reviewers 1 and 2 for providing insightful comments on our manuscript. We have addressed all the comments, suggestions and concerns raised by the reviewers and incorporated associated modifications in the manuscript. Reviewers' comments (in black font) and Authors' responses (in blue regular font) are given below. Texts in the manuscript are given in *blue italic font*.

**RC1: Comments from Reviewer 1**

**RC2: Comments from Reviewer 2**

**AC: Response by All Authors**
* * *
The manuscript by Ravi et al. deals with the estimation of $CO_2$ fluxes in India using a combination of model and vegetation-related remote sensing data. I think the study has some interesting points, such as the exploitation of SIF data to improve model-based estimate of carbon fluxes and the analysis of the spatio-temporal variation of these fluxes in India. The text is overall well written and the figures are clear.

**AC:** Thank you for your thorough review, helping us to improve the manuscript.

On the other hand, I have a number of methodological issues regarding the way in which SIF is used to improve GPP estimates by the model (this is the part that I can better cover):

**AC:** We have addressed all your comments/suggestions and revised the manuscript accordingly. Please see our responses below.

**RC1:**

- SIF products: two SIF products are chosen, one (TROPOSIF) is a real SIF product based on TROPOMI, whereas the other (GOSIF) is a merge of OCO-2 SIF retrievals and MODIS reflectance. TROPOSIF data have a coarser spatial resolution and a shorter time series than GOSIF, but a better temporal resolution and supposedly a higher sensitivity to vegetation physiological processes. I haven't been able to understand why the two products are chosen, since I don't see the synergies between the two exploited in the study (only a comparison in Fig.5).

**AC:** The two SIF products, GOSIF and TROPOSIF, were chosen to conduct a comparative examination of SIF products, exploit the SIF-GPP relationship in Indian biomes for utilizing them in the model as well as to cover the period from 2012 to 2020. While the longer time series in GOSIF benefits the study for covering the entire analysis period, better temporal resolution and (possible) higher sensitivity of TROPOSIF to vegetation processes (as pointed out by RC1) is expected to improve the model predictions of terrestrial biosphere flux dynamics across biomes. We used SIF-based GPP by GOSIF and TROPOSIF to modify the VPRM GPP simulations and then these simulations are evaluated with the Eddy Covariance (EC) observations. Based on the evaluation, we find that the model using TROPOSIF (VPRM$_{TROPOSIF, SMST}$) performs better than other model versions, and the rest of the analysis (from Sect. 3.4) utilized VPRM$_{TROPOSIF, SMST}$ simulations to understand the terrestrial biosphere flux dynamics. To make this clear, we have modified the statement as follows:

*L259-276: "We use two SIF products: GOSIF_v2, with longer data record (http://data.globalecology.unh.edu/; Li & Xiao (2019a)), and the TROPOMI-based product TROPOSIF, comparatively new product but with denser data coverage than GOSIF (http://ftp.sron.nl/open-access-data-2/TROPOMI/tropomi/sif/v2.1/l2b/; Guanter et al. (2021)). Given the scarcity of ground-based observational data and the potential of SIF as a proxy for deriving GPP, these publicly available SIF products are chosen to investigate the SIF-GPP relationship over India for the period covering from 2012 to 2020. A comparison of these two SIF products over India is made to examine any considerable product differences it may lead to variations in GPP derivation. GOSIF_v2 (hereafter referred to as GOSIF) provides SIF retrievals at spatial and temporal resolutions of 0.05° and 8-day. The spatial discontinuity in the original daily OCO-2 retrievals is improved in GOSIF using a machine learning approach based on MERRA-2 meteorological fields, MODIS reflectance, and land cover data, preserving the observed variability of discrete SIF retrievals as explained in Li & Xiao (2019a). In addition to SIF products, we also use the GPP product derived from OCO-2 SIF (Li & Xiao, 2019b), namely GOSIF_GPP_v2, providing 8-day GPP at 0.05° grid resolution for the model comparison (see details below). Daily SIF retrievals have been available from TROPOMI (hereafter referred to as TROPOSIF) since May 2018. We used the L2B data (Guanter et al., 2021) corrected for clouds (using retrievals with reflectance from cloud fraction less than 0.2) and gridded to 0.1°×0.1°."*

**RC1:**

- SIF-GPP scaling: actually, is there a real need for SIF products to improve VPRM GPP estimates? At the moment, SIF data are scaled to GPP using a 3[rd] SIF based product (GOSIF-GPP) as the GPP reference, and the resulting GPP(SIF) is used to scale the VPRM GPP output. I wonder, could one just directly link GOSIF-GPP to VPRM GPP for this scaling, without going through the separate SIF products as an intermediate step?

I think the authors should more clearly justify the use of the two SIF products, or move to a framework in which only GOSIF-GPP is used if there was no added-value in the use of the separate SIF products.

**AC:** Please see our response above. In addition to that please also see the following. This study has explored the usefulness of SIF in GPP estimations. We derived TROPOSIF-GPP by establishing the SIF-GPP relationship following Li & Xiao (2019b) across different biomes over India, as mentioned in Sect.2.2 (see Table 5). Our derived scalars (for converting SIF to GPP) are slightly different from Li & Xiao (2019b) due to the differences in Indian biomes, their classifications, and the upscaling of GOSIF products (see Table 5). Using two SIF products and VPRM, we have the following four products for ecosystem productivity: (SIF-derived) $GPP_{GOSIF}$, (SIF-derived) $GPP_{TROPOSIF}$, (VPRM-derived) $GPP_{GOSIF}$, and (VPRM-derived) $GPP_{TROPOSIF}$. We find that SIF-derived GPP products are closer to observations than uncalibrated (standard) VPRM (i.e., $VPRM_{STD}$) in terms of magnitude; however, the observed patterns in GPP are better captured by $VPRM_{STD}$ than SIF-derived products. The better performance of VPRM-derived GPP products indicates the potential of MODIS spectral reflectance bands (as used in VPRM) in determining vegetation dynamics and functioning, thereby on GPP estimates. This justifies the need to calibrate VPRM model parameters rather than simply use $GPP_{GOSIF}$ and $GPP_{TROPOSIF}$ in our NEE estimations. After model calibration with SIF products, the bias is reduced significantly (RMSE: $VPRM_{GOSIF}$= 4.9 µmol m$^{-2}$ s$^{-1}$, and $VPRM_{TROPOSIF}$= 4.3 µmol m$^{-2}$ s$^{-1}$ and MBE: $VPRM_{GOSIF}$ = -3.3 µmol m$^{-2}$ s$^{-1}$, $VPRM_{TROPOSIF}$= -2.6 µmol m$^{-2}$ s$^{-1}$). The manuscript is revised as follows to make it clear.

*L242-244: "The MODIS reflectance products used in VPRM captured vegetation dynamics over diverse Indian biomes better than SIF but failed to capture the magnitude when compared to observations."*

**RC1:**

Other comments:

Title: I would remove the list of satellites, as "CO2 fluxes of India derived from satellite observations"

AC: The new title can be: "Satellite-based sun-induced chlorophyll fluorescence and spectral reflectance improve the terrestrial biospheric $CO_2$ flux estimates in India."

**RC1:**

Abstract: I think it is too long and would greatly benefit from shortening.

AC: Done

**RC1:**

L122: I would say that current SIF retrievals actually suffer from low precision (high noise) rather than from systematic errors

AC: We modified the manuscript as:

*L123-126: "SIF retrievals are prone to various errors such as those associated with the strength, and extraction range of the signal, leaf scattering, re-absorption effects, and large background noise (Joiner et al., 2016; Köhler et al., 2015; Li et al., 2018; Liu et al., 2020)"*

**RC1:**

L123: the discussion on when and where SIF can be related to GPP (high light conditions etc) is important, and I think it should be extended.

AC: We modified the manuscript as:

*L126-132: "Various studies have demonstrated both linear and nonlinear SIF-GPP relationships (Guanter et al., 2014; Sun et al., 2018; Li et al., 2018; Kim et al., 2021). Environmental variables such as moisture content, temperature, radiation, and precipitation pattern, as well as measurement characteristics, such as SIF observation wavelength and angle, can affect the relationship between SIF and GPP (Wang et al., 2020; Chen et al., 2021;*

*Paul-Limoges et al., 2018; Guanter et al., 2012). Other factors, including plant functional types (PFTs) and plant physiology, also affect the SIF-GPP relationship (Sun et al., 2018)."*

**RC1:**

L128: Frankenberg et al. (2011) is a reference for GOSAT SIF, and Joiner et al., (2013) should be the one for GOME-2

**AC:** Corrected

**RC1:**

L134-154: reads more as Methods than as Introduction

**AC:** Modified. Please see L142-155 in the revised manuscript.

**RC1:**

L159: could you discuss how representative that one flux tower is for all the ecosystems in India? And is the tower footprint wide enough to allow comparison to 0.05 or 0.1° data?

**AC:** We modified the manuscript as:

*L364-367: "Generally, the flux towers have small footprints of around 1 km. The forest in which the flux tower is located covers an area of 176 ha and has a tree density of 400-500 trees/ha. The forest is homogeneous and free of anthropogenic impacts within a 1 km radius around the flux tower."*

*L377-382: "While model comparison with flux observations provides valuable insights into model performance, the interpretation needs to be done cautiously due to scale mismatches between the model and the flux observations. The future availability of more flux observations representing diverse biomes would enable us to perform a rigorous model evaluation at the ecosystem level, assessing errors due to model parametrization, inaccurate forcing data, and inadequate representation of ecosystem processes in the model."*

**RC1:**

L236: The reference for TROPOSIF is Guanter et al. (2021) (Koehler et al. would be for the Caltech SIF product)

**AC:** Corrected

**RC1:**

L243: TROPOSIF is daily, not hourly;

**AC:** Corrected

**RC1:**

also, how are the TROPOSIF data being used? Cloud fraction? Wavelength? Daylength-corrected or not? All these things really matter, especially in the frequently cloud-covered regions in India

**AC:** The analysis included only spectral reflectance data obtained under a cloud percentage of less than 0.2, as recommended by Guanter et al. (2021). We used daily TROPOSIF L2B product (http://ftp.sron.nl/open-access-data-2/TROPOMI/tropomi/sif/) and gridded to 0.1°×0.1° spatial grids.

We modified the manuscript as:

*L273-276: "Daily SIF retrievals have been available from TROPOMI (hereafter referred to as TROPOSIF) since May 2018. We used the L2B data (Guanter et al., 2021) corrected for clouds (using retrievals with reflectance from cloud fraction less than 0.2) and gridded to 0.1°×0.1°."*

**RC1:**

L346: I actually find it surprising how low these correlations are (perhaps only due to random noise?)

**AC:** Studies have shown that the SIF-GPP relationship varies with factors such as vegetation type, season, and meteorological conditions (Yang et al., 2017; Li et al., 2018; Guanter et al., 2012). Leaf physiology is also important in determining the SIF-GPP relationship (Wu et al., 2022).

**RC1:**

L362: only 743-758 nm retrievals should be used for TROPOSIF, there are issues with 735-758 nm retrievals (see Guanter et al., 2021)

**AC:** Thank you for noting this. The range is mistakenly given in the manuscript text. We have used 743-758 nm retrievals for the analysis. The manuscript is revised as follows:

*L413-416: "Overall, we find that TROPOSIF values (based on SIF retrievals at 743-758 nm fitting window) are ~4 times greater than GOSIF (based on SIF retrievals at 757 nm) over the study region for all the biomes except for Grassland, where the biome-specific TROPOSIF is ~3 times larger than GOSIF."*

**RC1:**

L374: the double growing season and the impact of climate on SIF in India are also discussed in https://doi.org/10.1073/pnas.1320008111 and https://doi.org/10.1111/gcb.14302

**AC:** We modified the manuscript as:

*L435-436: "Song et al. (2018) and Guanter et al. (2014) explored the ability of SIF to capture the double growing season of crops, as well as the impact of climate on SIF in India."*

**RC2: Reviewer 2**

This is a review of "Spatiotemporal variations in terrestrial biospheric CO2 fluxes of India derived from MODIS, OCO-2 and TROPOMI satellite observations and a diagnostic terrestrial vegetation model" by Ravi, et al., under consideration for publication in Biogeosciences.

The article presents a novel method to estimate regional gross primary productivity (GPP), ecosystem respiration (Re), and net ecosytem exchange (NEE) in regions sparsely covered by eddy covariance (EC) observations. This topic is important, as large areas of the world's land mass are not well characterised by EC observations. The authors begin with an ecosystem model, VPRM, with an extensive history of peer-reviewed publications documenting its application to regional flux estimation. Traditionally VPRM would be calibrated against EC observations; the authors address the sparsity of EC sites in India by nudging VPRM's fluxes toward solar-induced fluorescence (SIF)-derived GPP and Re derived

from the FLUXCOM global analysis of eddy covariance datasets. This approach is, to my knowledge, novel and in my opinion worthwhile. The paper is well-written. I outline below some questions and concerns I have; in my opinion the article may be published in Biogeosciences if these are addressed.

AC: Thank you for your thorough review. All your comments are addressed, which helped us to improve the manuscript.

CONCEPTUAL                    QUESTIONS                    AND                    CONCERNS
==================================

This study has presented a novel way of estimating GPP, Re, and NEE via coupling SIF observations, FLUXCOM/FLUXNET, and VPRM. These new flux estimates are *different* from previous estimates; In my opinion, ideally this study should answer the overarching question "are these new flux estimates *better* than existing methods". The authors demonstrate that their results fit the Betul EC dataset better than the TRENDYv10 ensemble or the individual driver datasets (SIF, FLUXCOM, FLUXNET) used. However, I am somewhat perplexed by the methodology described in sections 2.2 and 2.3. I am concerned that the assessment of model improvement relies on improved correlation at a single EC site (Betul) with no assessment of goodness of fit versus parsimony relative to VPRMstd or discussion of how representative Betul is of India.

It makes sense to me to use other datasets (SIF, FLUXCOM/FLUXNET) to drive VPRM in the absence of EC data. But eq (8) makes the modified GPP a linear function of SIF-derived GPP. This raises several concerns. First, it makes sense to me that GPPvprm,mod fits the Betul data better than the GPPvprm alone, because GPPvprm,mod introduces additional parameters to the model. I think it is necessary to some sort of fit-parsimony calculation (e.g. Akaike's Information Criterion, Bayesian Information Criterion, etc.) to assess whether VPRMrefined truly improves VPRMstd.

AC: SIF is used to calibrate VPRM GPP parameters. The number of model parameters remains the same in the case of GPP. But we have done parameter addition in VPRM $R_{eco}$. The Bayesian Information Criterion (BIC) analysis conducted among the VPRM$_{STD}$ ($-1.86 \times 10^6$) and VPRM$_{refined}$ provided the lowest BIC value for VPRM$_{refined}$ ($-6.65 \times 10^6$).

We have revised the manuscript as:

*L555-557: "This was further corroborated by the statistical comparison of VPRM$_{refined}$ and VPRM$_{STD}$ using the Bayesian Information Criterion (BIC), which showed that VPRM$_{refined}$ (-6.65×10$^6$) had a lower BIC value than VPRM$_{STD}$ (-1.86×10$^6$)."*

**RC2:**

Second, did you consider estimating VPRMstd parameter values by minimizing its difference with the SIF-derived GPP products over your domain? This seems a more rigorous approach to me.

**AC:** The objective of this study is not to calibrate the VPRM$_{STD}$ parameters using existing GPP products (as truth) but rather to refine the model using information from SIF in addition to MODIS reflectance data. Please note that the VPRM$_{STD}$ GPP captured the seasonality better than SIF-derived GPP products but with a large offset in magnitude.

Please see lines: *L242-244: "The MODIS reflectance products used in VPRM GPP estimation captured vegetation dynamics over diverse Indian biomes better than SIF but failed to capture the magnitude when compared to observations."*

**RC2:**

Betul is not the only EC site in India: see also Barkot Flux Research Site, IARI Flux Site, Haldwani Forest Plantation, all operating since 2012. IARI Flux Site seems of particular relevance, as it observes a cropland and croplands comprise almost 70 percent of India's land area (L557). Did you consider estimating parameters for VPRMstd by minimising the VPRMstd error against all these EC sites? Seems to me you would then have a much better starting point than those estimated against Amazonian biomes (L225). You could build in SIF to the VPRM GPP equation (eq 8), estimate the SIF, LSWI, and PAR parameters jointly, and test against held-out EC data. I think some text is needed to justify the current setup of the study.

**AC:** We agree that more EC observations would benefit model calibration or evaluation. However, the measurements mentioned above are not publicly available (those sites come under the ASIAFLUX network, https://db.cger.nies.go.jp/asiafluxdb/?page_id=43). We are

actively making contacts with data owners across Indian subcontinent and attempting to support common flux database services. Hopefully, the situation will improve in future studies. In addition to this, please also see our response to comments below.

We have revised the manuscript as:

*L379-382: "The future availability of more flux observations representing diverse biomes would enable us to perform a rigorous model evaluation at the ecosystem level, assessing errors due to model parametrization, inaccurate forcing data and inadequate representation of ecosystem processes in the model."*

*L234-237: "Due to the lack of availability of sufficient observational eddy flux measurements for calibration for India, we use the VPRM parameters that were optimised against the Amazonian Tropical biomes (Botía et al., 2022) as given in Table 2."*

The following text is already there in the manuscript:
*L740-742: "The increased number of flux tower observations in the future will help to optimise the model parameters to enhance the robustness of these simulations."*

**RC2:**

When calibrating Reco,vprm,mod parameters (L278-283), did you remove Betul's data from the calibration dataset? This is important because you are evaluating model performance against Betul. How did you calibrate the parameters? You must describe this -- there are entire papers on this topic, and there is no description here.

**AC:** Betul data is not part of the calibration.

Please see *L322-326 "The terrestrial vegetation fluxes (specifically ecosystem respiration fluxes) derived from 1) FLUXNET (https://db.cger.nies.go.jp/DL/10.17595/20200227.001.html.en, see Table 1, Zeng, Jiye (2020)) and 2) FLUXCOM (https://www.bgc-jena.mpg.de/geodb/projects/DataDnld.php, see Table 1, Jung et al. (2020)) observational database are used for parameter optimization."*

Also, we have now included a flowchart to make it clear. Please see Figure 2 in the manuscript.

**RC2:**

FLUXCOM and FLUXNET are global datasets. What sites did you use when calibrating the respiration model (L283 to 288)? Are those sites in India? If not, why do you think they will work better than or are more appropriate than, say, the Amazonian parameters which you note (L227) might lead to reduced model performance?

**AC:** The EC site observations would have been an ideal choice to start with the model parameter optimization. However, the scarcity of publicly available EC observations in India limits the site-based model calibration. Hence, we have chosen FLUXCOM and FLUXNET global databases for the respiration calibration. The model calibration is done separately for each vegetation class by considering all respiration values corresponding to each vegetation class in our domain. While SIF retrievals provide vital information to GPP calibration, the scarcity of observational-based evidence can make the ecosystem respiration fluxes less reliable. Hence, the calibration we have done for respiration in the absence of EC observations may not be enough, which can lead to systematic biases in our NEE estimations. A potential future step would be to combine atmospheric data and VPRM through inverse modelling to constrain the carbon balance better.

To clarify this, we have revised the manuscript as follows:

*L326-330: "We have chosen the above global datasets for calibration of $R_{eco}$ due to the lack of sufficient publicly available EC observations in India. The model calibration is done separately for each vegetation class by considering all respiration fluxes corresponding to each vegetation class in our domain. Table 2 provides the details of the vegetation-specific model parameters derived for refining $R_{eco}$."*

SPECIFIC                             QUESTIONS                             AND                             CONCERNS

==============================

**RC2:**

This study uses a great many datasets for different phases of model tuning and evaluation. I think the article would benefit greatly from a flowchart-style figure early in the text summarizing which datasets informed which pieces of the process.

AC: Thank you for the suggestion. We have now included a flowchart in the manuscript. Please see Figure 2 in the manuscript.

**RC2:**

L226: what modifications did you make to these parameters, and why? Please make this explicit.

**AC:** We modified the manuscript as:

*L234-237: "Due to the lack of availability of sufficient observational eddy flux measurements for calibration for India, we use the VPRM parameters that were optimised against the Amazonian Tropical biomes (Botía et al., 2022) as given in Table 2."*

**RC2:**

How good is the GPP-SIF relationship in India? These lines note that the relationship weakens during drought stress, and India has a distinct wet/dry season.

**AC:** Thank you for this question. The SIF-GPP relationship varies with environmental factors such as moisture content, temperature, radiation, precipitation pattern, and measurement characteristics such as SIF observation wavelength and angle. Also, plant functional type affects the relationship. Despite the importance, not enough studies demonstrate the SIF-GPP relationships across biomes in India under different meteorological/plant physiological conditions. We do think that future study(ies) is (are) required to investigate in detail, given the availability of EC measurements and to establish these (linear/non-linear) relationships in different seasons across India.

The manuscript is revised as follows:

*L123-132: "SIF retrievals are prone to various errors such as those associated with the strength, and extraction range of the signal, leaf scattering, re-absorption effects, and large background noise (Joiner et al., 2016; Köhler et al., 2015; Li et al., 2018; Liu et al., 2020). Various studies have demonstrated both linear and nonlinear SIF-GPP relationships (Guanter et al., 2014; Sun et al., 2018; Li et al., 2018; Kim et al., 2021). Environmental variables such as moisture content, temperature, radiation, and precipitation pattern as well as measurement*

*characteristics such as SIF observation wavelength and angle can affect the relationship between SIF and GPP (Wang et al., 2020; Chen et al., 2021; Paul-Limoges et al., 2018; Guanter et al., 2012). Other factors, including plant functional types (PFTs) and plant physiology, also affect the SIF-GPP relationship (Sun et al., 2018)."*

*L449-452: "It is noteworthy that the SIF-GPP relationship can become weak in certain environmental conditions such as drought (e.g., Shekhar et al. (2022) and be variable within certain biome based on leaf physiology (e.g., Wu et al. (2022)). However, a future study is needed to elucidate SIF-GPP relationships in India across different biomes in drought/wet conditions."*

**RC2:**

L260: The sigma indicates a summation; what are you summing over? Also, what is epsilon? I assume some sort of error term, but it is not defined. Please define it and explain how you determined its value.

**AC:** The manuscript is revised as follows:

*L296-300:*

$$\eta_{vg} = \frac{\sum_{i=1}^{n1} \sum_{j=1}^{n2} \sum_{t=1}^{n3} (GPP_{SIF}(i,j,t,vg) \times GPP_{vprm,STD}(i,j,t,vg))}{\sum_{i=1}^{n1} \sum_{j=1}^{n2} \sum_{t=1}^{n3} GPP_{vprm,STD}(i,j,t,vg)^2}$$

*$\varepsilon$ in Eq (8) represents the vegetation specific error term or the y intercept between $GPP_{SIF}(vg)$ and $GPP_{vprm,STD}(vg)$. n1, n2, and n3 represents the number of latitude, longitude, and time indices per vegetation class.*

**RC2:**

L264: How do you derive or obtain this relationship between GPPsif and GPPvprm?

**AC:** Please see the modified equation 9.

**RC2:**

L350: "the highest SIF values": are you talking about SIF, or SIF-derived GPP? SIF and GPP are inversely correlated; if the desert areas have the lowest values I presume you are talking about SIF-derived GPP, not SIF itself.

**AC:** SIF and GPP are positively correlated - please note the sign used (e.g., Gao et al. 2021, Yang et al., 2017; Li et al., 2018; Guanter et al., 2012).

**RC2:**

L389: I'm confused; GPP and SIF are inversely correlated, but these scalars describe a linear relationship with positive slope.

**AC:** Please see our response above and also see Eq (1) in the manuscript.

**RC2:**

L659-661: Please remove this sentence - it is a tautology. VPRM is driven by temperature, moisture, and radiation; it follows that its GPP, Re, and NEE spatial heterogeneity must vary with those drivers!

**AC:** Done

**RC2:**

Is the code used for analysis and plotting publicly available?

**AC:** Not the analysis/plotting codes are publicly available.

**RC2:**

Table 2 is confusing: Do the 'a.' and 'b.' in the caption correspond to the aT, bT, aM, bM, etc? If so, the different typesetting (boldface in caption, italics in table) is confusing. If not, what do a and b from the caption refer to in the table? There is a "2" floating in space between the Savanna and Shrubland lines, and two "6"s floating in space below the Mixed Forest line.

**AC:** Corrected

**RC2:**

Fig. 1: The colormap makes it very hard to distinguish Savanna, Grassland, Shrubland in the map, and deciduous forest from mixed. Please consider higher-contrast colors.

**AC:** Corrected

**RC2:**

Fig 3: Are the GOSIF values in the first row scaled up as described in L361-365? The GOSIF values in the plot do not appear 3 to 4 times the TROPOSIF values. If the plot shows the adjusted values, please note in the caption.

**AC:** Corrected

**RC2:**

Fig 5: please show VPRMrefined in this figure. VPRMrefined is your main product, yes?

**AC:** Yes, $VPRM_{refined}$ is our main product, and it is shown in the figure. $VPRM_{TROPOSIF, SMST}$ is renamed as $VPRM_{refined}$ (pl see lines 553-555)

TECHNICAL          CORRECTIONS          AND          TYPOGRAPHICAL          ERRORS

============================================

**RC2:**

L77: change "their net" to "their net difference"

**AC**: Done.

**RC2:**

L88-89: "its importance in the global carbon budget": citation needed

**AC**: Done. The manuscript is revised as follows:

*L85-88: "For example, the models are constrained with few observations over the Indian subcontinent, resulting in low confidence in the estimates of fluxes over India despite its important role in the global carbon budget (Thompson et al., 2016)."*

**RC2:**

L105: "coarse resolution, e.g., 2' x 2'" - did the authors really mean two minutes? 1/30 degrees by 1/30 degrees is much higher resolution than anything in this study or most others. Did the authors mean "2° x 2° (that is, 2 degrees by 2 degrees)?

**AC**: Corrected

The manuscript is revised as follows:

*L104-108: "However, these models are employed at coarse resolution, e.g., monthly temporal resolution for CASA (but with higher spatial resolution), and TRENDY with sub-daily temporal resolution (with output available monthly) and varying spatial resolution with respect to the model, typical 0.5° or above (see Table 3 for further details), with limited model validation against observations over India."*

**RC2:**

L148: "We expect...": citation needed

**AC**: Done (The sentence is removed)

**RC2:**

L346: change "regirdded" to "regridded"

**AC**: Done

**RC2:**

L354: "(2019 to 2020)": please remove the parentheses

**AC**: Done

**RC2:**

L530: "uptake capacity of the Indian region by -0.14 Pg" - is there a word missing between "region" and "by"?  Please reword.

**AC**: The manuscript is revised as follows:

*L589-590: "A similar study using TRENDY models by Rao et al. (2019) also showed the uptake capacity of the Indian region  (-0.14 Pg C yr$^{-1}$ from 1901 to 2010)."*

 **RC2:**

L563: "The highest productivity of forest ecosystems over Grassland": This confuses me - please reword.

**AC**: The manuscript is revised as follows:

*L623-624: "The highest productivity of ecosystems dominated by forest over Grassland is also seen in other parts of the globe (e.g., Yu et al., 2013)."*

**RC2:**

L671: "when the respiration model parameters calibrated using FLUXNET": parameters ARE/WERE calibrated?

**AC**: The manuscript is revised as follows:

*L730-733: "Overall, we find that the Indian biosphere acts as a sink with an annual NEE ranging from -0.38 Pg C yr$^{-1}$ (-0.51 Pg C yr$^{-1}$) to -0.53 Pg C yr$^{-1}$ (-0.88 Pg C yr$^{-1}$) when the respiration model parameters were calibrated using FLUXNET (FLUXCOM) and an annual GPP ranging 3.39 yr$^{-1}$ to 3.88 Pg C yr$^{-1}$ for the years from 2012 to 2020."*

**RC2:**

Fig 5: Is the solid black line in the legend correspond to the dashed black line in the plot?

**AC:** Yes, the figure is modified.

**RC2:**

Fig 10, 11: These colormaps make it very difficult to resolve one line from one another in these two plots. Please consider dots, dashes, different plot markers to help distinguish.

**AC:** The figures are modified based on the suggestion.

References:

Botía, S., Komiya, S., Marshall, J., Koch, T., Gałkowski, M., Lavric, J., Gomes-Alves, E., Walter, D., Fisch, G., Pinho, D. M., Nelson, B. W., Martins, G., Luijkx, I. T., Koren, G., Florentie, L., Carioca De Araújo, A., Sá, M., Andreae, M. O., Heimann, M., Peters, W., and Gerbig, C.: The $CO_2$ record at the Amazon Tall Tower Observatory: A new opportunity to study processes on seasonal and inter-annual scales, Glob. Change Biol., 28, 588–611, https://doi.org/10.1111/gcb.15905, 2022.

Chen, A., Mao, J., Ricciuto, D., Lu, D., Xiao, J., Li, X., Thornton, P. E., and Knapp, A. K.: Seasonal changes in GPP/SIF ratios and their climatic determinants across the Northern Hemisphere, Glob. Change Biol., 27, 5186–5197, https://doi.org/10.1111/gcb.15775, 2021.

Frankenberg, C., Fisher, J. B., Worden, J., Badgley, G., Saatchi, S. S., Lee, J.-E., Toon, G. C., Butz, A., Jung, M., Kuze, A., and Yokota, T.: New global observations of the terrestrial carbon cycle from GOSAT: Patterns of plant fluorescence with gross primary productivity: CHLOROPHYLL FLUORESCENCE FROM SPACE, Geophys. Res. Lett., 38, n/a-n/a, https://doi.org/10.1029/2011GL048738, 2011.

Gao, H., Liu, S., Lu, W., Smith, A. R., Valbuena, R., Yan, W., Wang, Z., Xiao, L., Peng, X., Li, Q., Feng, Y., McDonald, M., Pagella, T., Liao, J., Wu, Z., and Zhang, G.: Global Analysis of the Relationship between Reconstructed Solar-Induced Chlorophyll Fluorescence (SIF) and Gross Primary Production (GPP), Remote Sens., 13, 2824, https://doi.org/10.3390/rs13142824, 2021.

Guanter, L., Frankenberg, C., Dudhia, A., Lewis, P. E., Gómez-Dans, J., Kuze, A., Suto, H., and Grainger, R. G.: Retrieval and global assessment of terrestrial chlorophyll fluorescence from GOSAT space measurements, Remote Sens. Environ., 121, 236–251,

https://doi.org/10.1016/j.rse.2012.02.006, 2012.

Guanter, L., Zhang, Y., Jung, M., Joiner, J., Voigt, M., Berry, J. A., Frankenberg, C., Huete, A. R., Zarco-Tejada, P., Lee, J.-E., Moran, M. S., Ponce-Campos, G., Beer, C., Camps-Valls, G., Buchmann, N., Gianelle, D., Klumpp, K., Cescatti, A., Baker, J. M., and Griffis, T. J.: Global and time-resolved monitoring of crop photosynthesis with chlorophyll fluorescence, Proc. Natl. Acad. Sci., 111, https://doi.org/10.1073/pnas.1320008111, 2014.

Guanter, L., Bacour, C., Schneider, A., Aben, I., van Kempen, T. A., Maignan, F., Retscher, C., Köhler, P., Frankenberg, C., Joiner, J., and Zhang, Y.: The TROPOSIF global sun-induced fluorescence dataset from the Sentinel-5P TROPOMI mission, Earth Syst. Sci. Data, 13, 5423–5440, https://doi.org/10.5194/essd-13-5423-2021, 2021.

Joiner, J., Guanter, L., Lindstrot, R., Voigt, M., Vasilkov, A. P., Middleton, E. M., Huemmrich, K. F., Yoshida, Y., and Frankenberg, C.: Global monitoring of terrestrial chlorophyll fluorescence from moderate-spectral-resolution near-infrared satellite measurements: methodology, simulations, and application to GOME-2, Atmospheric Meas. Tech., 6, 2803–2823, https://doi.org/10.5194/amt-6-2803-2013, 2013.

Joiner, J., Yoshida, Y., Guanter, L., and Middleton, E. M.: New methods for the retrieval of chlorophyll red fluorescence from hyperspectral satellite instruments: simulations andapplication to GOME-2 and SCIAMACHY, Atmospheric Meas. Tech., 9, 3939–3967, https://doi.org/10.5194/amt-9-3939-2016, 2016.

Jung, M., Schwalm, C., Migliavacca, M., Walther, S., Camps-Valls, G., Koirala, S., Anthoni, P., Besnard, S., Bodesheim, P., Carvalhais, N., Chevallier, F., Gans, F., Goll, D. S., Haverd, V., Köhler, P., Ichii, K., Jain, A. K., Liu, J., Lombardozzi, D., Nabel, J. E. M. S., Nelson, J. A., O'Sullivan, M., Pallandt, M., Papale, D., Peters, W., Pongratz, J., Rödenbeck, C., Sitch, S., Tramontana, G., Walker, A., Weber, U., and Reichstein, M.: Scaling carbon fluxes from eddy covariance sites to globe: synthesis and evaluation of the FLUXCOM approach, Biogeosciences, 17, 1343–1365, https://doi.org/10.5194/bg-17-1343-2020, 2020.

Kim, J., Ryu, Y., Dechant, B., Lee, H., Kim, H. S., Kornfeld, A., and Berry, J. A.: Solar-induced chlorophyll fluorescence is non-linearly related to canopy photosynthesis in a temperate evergreen needleleaf forest during the fall transition, Remote Sens. Environ., 258, 112362, https://doi.org/10.1016/j.rse.2021.112362, 2021.

Köhler, P., Guanter, L., and Joiner, J.: A linear method for the retrieval of sun-induced chlorophyll fluorescence from GOME-2 and SCIAMACHY data, Atmospheric Meas. Tech., 8, 2589–2608, https://doi.org/10.5194/amt-8-2589-2015, 2015.

Li and Xiao: Mapping Photosynthesis Solely from Solar-Induced Chlorophyll Fluorescence:

A Global, Fine-Resolution Dataset of Gross Primary Production Derived from OCO-2, Remote Sens., 11, 2563, https://doi.org/10.3390/rs11212563, 2019a.

Li, X. and Xiao, J.: A Global, 0.05-Degree Product of Solar-Induced Chlorophyll Fluorescence Derived from OCO-2, MODIS, and Reanalysis Data, Remote Sens., 11, 517, https://doi.org/10.3390/rs11050517, 2019b.

Li, X., Xiao, J., and He, B.: Chlorophyll fluorescence observed by OCO-2 is strongly related to gross primary productivity estimated from flux towers in temperate forests, Remote Sens. Environ., 204, 659–671, https://doi.org/10.1016/j.rse.2017.09.034, 2018.

Liu, X., Liu, L., Hu, J., Guo, J., and Du, S.: Improving the potential of red SIF for estimating GPP by downscaling from the canopy level to the photosystem level, Agric. For. Meteorol., 281, 107846, https://doi.org/10.1016/j.agrformet.2019.107846, 2020.

Paul-Limoges, E., Damm, A., Hueni, A., Liebisch, F., Eugster, W., Schaepman, M. E., and Buchmann, N.: Effect of environmental conditions on sun-induced fluorescence in a mixed forest and a cropland, Remote Sens. Environ., 219, 310–323, https://doi.org/10.1016/j.rse.2018.10.018, 2018.

Shekhar, A., Buchmann, N., and Gharun, M.: How well do recently reconstructed solar-induced fluorescence datasets model gross primary productivity?, Remote Sens. Environ., 283, 113282, https://doi.org/10.1016/j.rse.2022.113282, 2022.

Song, L., Guanter, L., Guan, K., You, L., Huete, A., Ju, W., and Zhang, Y.: Satellite sun-induced chlorophyll fluorescence detects early response of winter wheat to heat stress in the Indian Indo-Gangetic Plains, Glob. Change Biol., 24, 4023–4037, https://doi.org/10.1111/gcb.14302, 2018.

Sun, Y., Frankenberg, C., Jung, M., Joiner, J., Guanter, L., Köhler, P., and Magney, T.: Overview of Solar-Induced chlorophyll Fluorescence (SIF) from the Orbiting Carbon Observatory-2: Retrieval, cross-mission comparison, and global monitoring for GPP, Remote Sens. Environ., 209, 808–823, https://doi.org/10.1016/j.rse.2018.02.016, 2018.

Wang, X., Chen, J. M., and Ju, W.: Photochemical reflectance index (PRI) can be used to improve the relationship between gross primary productivity (GPP) and sun-induced chlorophyll fluorescence (SIF), Remote Sens. Environ., 246, 111888, https://doi.org/10.1016/j.rse.2020.111888, 2020.

Wu, G., Guan, K., Jiang, C., Kimm, H., Miao, G., Bernacchi, C. J., Moore, C. E., Ainsworth, E. A., Yang, X., Berry, J. A., Frankenberg, C., and Chen, M.: Attributing differences of solar-induced chlorophyll fluorescence (SIF)-gross primary production (GPP) relationships between two C4 crops: corn and miscanthus, Agric. For. Meteorol., 323, 109046,

https://doi.org/10.1016/j.agrformet.2022.109046, 2022.

Yang, H., Yang, X., Zhang, Y., Heskel, M. A., Lu, X., Munger, J. W., Sun, S., and Tang, J.: Chlorophyll fluorescence tracks seasonal variations of photosynthesis from leaf to canopy in a temperate forest, Glob. Change Biol., 23, 2874–2886, https://doi.org/10.1111/gcb.13590, 2017.

Zeng, Jiye: A Data-driven Upscale Product of Global Gross Primary Production, Net Ecosystem Exchange and Ecosystem Respiration (ver.2020.2), https://doi.org/10.17595/20200227.001, 2020.